# Self-organization of in vitro neuronal assemblies drives to complex network topology

**Priscila C Antonello[1], Thomas F Varley[2,3], John Beggs[4], Marimélia Porcionatto[1], Olaf Sporns[2†], Jean Faber[5*†]**

[1]Department of Biochemistry – Escola Paulista de Medicina – Universidade Federal de São Paulo (UNIFESP), São Paulo, Brazil; [2]Department of Psychological and Brain Sciences, Indiana University, Bloomington, United States; [3]Department of Informatics, Computing, and Engineering, Indiana University, Bloomington, United States; [4]Department of Physics, Indiana University, Bloomington, United States; [5]Department of Neurology and Neurosurgery – Escola Paulista de Medicina – Universidade Federal de São Paulo (UNIFESP), São Paulo, Brazil

**\*For correspondence:**
pcantoneli@unifesp.br (PCA);
jean.faber@unifesp.br (JF)

†These authors contributed equally to this work

**Competing interest:** The authors declare that no competing interests exist.

**ABSTRACT** Activity-dependent self-organization plays an important role in the formation of specific and stereotyped connectivity patterns in neural circuits. By combining neuronal cultures, and tools with approaches from network neuroscience and information theory, we can study how complex network topology emerges from local neuronal interactions. We constructed effective connectivity networks using a transfer entropy analysis of spike trains recorded from rat embryo dissociated hippocampal neuron cultures between 6 and 35 days in vitro to investigate how the topology evolves during maturation. The methodology for constructing the networks considered the synapse delay and addressed the influence of firing rate and population bursts as well as spurious effects on the inference of connections. We found that the number of links in the networks grew over the course of development, shifting from a segregated to a more integrated architecture. As part of this progression, three significant aspects of complex network topology emerged. In agreement with previous in silico and in vitro studies, a small-world architecture was detected, largely due to strong clustering among neurons. Additionally, the networks developed in a modular topology, with most modules comprising nearby neurons. Finally, highly active neurons acquired topological characteristics that made them important nodes to the network and integrators of modules. These findings leverage new insights into how neuronal effective network topology relates to neuronal assembly self-organization mechanisms.

## Editor's evaluation

This paper investigates the emergence of complex network organization in neuronal circuits grown in vitro. Network analysis of neuronal activity recordings allowed a detailed assessment of how neurons self-organise into clusters of functionally segregated models while also retaining a capacity for integrated communication through a subset of highly active neurons. This work is of interest to researchers working on neuronal connectivity, brain development, and self-organisation in complex systems.

## Introduction

Self-organizing systems are those capable of changing their internal structure and/or function from specific interactions of their components, given an appropriate exchange of matter or energy with the environment (*Prokopenko, 2009*). In these systems, the interaction among components generates

stability against external noise, promoting physical conditions to the emergence of hierarchical structures and new functions (*Banzhaf, 2009*). Neural systems present all of these characteristics, where the interaction among neurons allows for the formation of complex patterns, suggesting that neural circuits are an example of a self-organizing system (*Isaeva, 2012*).

During neurodevelopment, synaptic connections are set up by activity- and non-activity-dependent factors (*Goodman and Shatz, 1993*). Initially, axonal outgrowth and neuronal migration are guided by chemical, mechanical, and geometric attractors/repellents for long distances, toward specific targets (*Goodman, 1996*; *SenGupta et al., 2021*). Once axons or neurons reach the set of biochemically targeted neurons, they develop an overall coarse-grained scaffold of connections (*Goodman and Shatz, 1993*). From that point, neuronal activity refines the connections in very specific patterns, tuning the network to process information (*Zhang and Poo, 2001*). Synapses promote the structural and functional coupling of neurons, by allowing the propagation of biochemical and electric signals (*Südhof, 2021*). However, to produce a significant post-synaptic effect the pre-synaptic stimuli need to follow some specific temporal and spatial requirements (*Magee, 2000*). Adaptative mechanisms regulate synapses by promoting cooperation and competition among them (*Zhang et al., 1998*). In this way, the maturation of synapses induced by neuronal spontaneous activity has an important role in the emergence of spatial and temporal patterns related to self-organizing systems.

However, the self-organization process happens under very specific environmental conditions, via mechanisms that are still not fully understood. For instance, how does neuronal activity influence these initial network structures, yielding specific configurations? Are there privileged network topologies that are 'locked-in' by neuronal intrinsic processes? What are the consequences of high or low heterogeneity of neuronal activity for the network configuration?

One way to address these questions is to model the development of neuronal assemblies as a network. Understanding how these networked architectures emerge from the interactions between neurons forming circuits may give us insights into factors that influence neuronal communication, information processing, and ultimately the emergence of neurodevelopmental disorders. Primary cultures of dissociated neural progenitors or immature neurons present spontaneous activity after a few days in vitro (DIV) and show dynamic changes over time as the neuronal assembly evolves (*Cohen et al., 2008*; *Maeda et al., 1995*; *Pasquale et al., 2008*). The ability of neurons to develop connections and complex dynamics in vitro provides a convenient approach to studying activity-dependent self-organization. Once these preparations represent a simpler model of neural circuits where neuron properties can be studied separately. *Downes, 2012* and *Schroeter et al., 2015* showed how complex topologies such as small-world architecture, modular organization, hubs, and rich-clubs emerge from functional networks. However, they considered a coarse-grained scale by using neuron populations as the fundamental element of the networks. Herein we propose to go deeper into these analyses by studying the emergence of complex network topology from networks where neurons are the main element, and the effective connectivity is the link between them. We computed effective connectivity networks using spike trains recorded from hippocampal neuronal cultures over the ~30 days of maturation. Additionally, we also considered neuronal firing rate and physical location to investigate how the self-organization of neuronal assembly relates to the network topology. Effective connectivity provides a measure of the influence one neuron exerts over another, to study the process in which effective networks organize into specific architectures may bring new perspectives on how neural circuits accommodate complex dynamics and functions.

Unlike other types of in vitro preparations, cultures of dissociated neurons display a consistent pattern of population bursts. Because many neurons fire synchronously during these bursts, it can be challenging to extract effective connections from the spiking data (*Ito et al., 2011*; *Penn et al., 2016*; *Wagenaar et al., 2006*). To address this issue as well as the bias caused by spurious connections, we applied a very stringent transfer entropy (TE) analysis (*Ito et al., 2011*; *Schreiber, 2000*) which we then validated using synthetic data. For details, see the Materials and methods section. Even with this conservative approach, the resulting effective networks showed growth in edge density, progressing from a more segregated to a more integrated architecture. These topological changes drove a combination of neuron clustering and five specifically directed 3-node motifs over time. The neuronal population was divided into modules of a few near but not necessarily adjacent neighbors. In these modular organizations, the most active neurons served as 'integrators,' connecting distinct modules. These

results demonstrate how network topology develops into segregated and integrated architectures and point toward a role for firing rate in regulating the emergence of complex network topologies.

## Results

We analyzed the temporal evolution of effective network topology during the development of dissociated hippocampal neuronal assemblies isolated from rats' embryos. The set of signals used here is from the freely available data set on the Collaborative Research in Computational Neuroscience (CRCNS) data sharing initiative (*Timme et al., 2016c*).

### In vitro effective networks

Effective connectivity was inferred by using TE analysis, which provided weighted and directed networks for each of the cultures. Only networks with the edge density within the 95% confidence interval were used. In this way 11 networks were removed from the total set, resulting in 424 networks, which we grouped into periods based on how long the cultures had been developing (6, 9, 12,15,18, 21, 24, 27, 30, 33, and 35 DIV). The total number of networks analyzed by DIV is shown in *Figure 1a* (min: 9, max: 18, mean: 15). *Figure 1b* shows a decrease in the number of recorded neurons over DIV that become more intense after 30 DIV.

Consistent with previous in vivo (*Song et al., 2005*), in vitro (*Nigam et al., 2016*; *Timme et al., 2016b*), and in silico studies (*Neymotin et al., 2011*) the neuron firing rate distribution (*Figure 1c*) and connection weight distribution (*Figure 1d*) for all DIV followed a log-normal distribution (p>0.01 for both analysis in all DIV). The left tail of the firing rate distributions is diminished due to neuron cutoffs with a firing rate lower than 0.2 Hz.

The degree distributions (*Figure 1e*) were consistent with a heavy-tailed distribution with slight variability over the DIV. A similar degree distribution was reported by *Pasquale et al., 2008* and *Timme et al., 2016b* using mature networks (after 14 DIV). The distributions shifted to higher values of degree as maturation progressed, indicating an increase in connectivity along with development (inset of *Figure 1e*). The same results were observed for the connection strength distributions (*Figure 1f*).

We found that the joint probability distributions of connection weights and distance between neurons changed until the 18 DIV, and then stabilized (*Figure 2a* and *Figure 2—figure supplement 1*). On 6–9 DIV, most of the connections were short (higher probability for 25 and 205 µm, p≤0.05, *Figure 2b*) and weighted around the distribution geometric mean (p≤0.05, *Figure 2c*). After 12 DIV the probability of longer connections increased, shifting the distributions to the right side. Only some connections extended for long distances (*Figure 2b*), and these connections were more likely to also have a weight close to the geometric mean (p≤0.05 only between 18 and 33 DIV, *Figure 2d*). *Figure 2e* shows the raise of the distance between neurons over DIV illustrating the gradual evolution from a segregated to an integrated network.

### Validation of effective connectivity

In our analysis, since we have access only to neuron spiking activity, the term 'connections' reflects strong paths for information flow rather than actual structural connections (*Lizier and Prokopenko, 2010*). This type of inference in neural systems should be done carefully because of the number of influences that the analysis can have. Additionally, the relationship between the structural and effective connections may not fully match. Since the predicted transfer is computed from spike trains, not all possible dynamical states are presented in the spontaneous activity. With inhibition active, several communication paths may be inactive or silent, although structural connections may exist (*Park and Friston, 2013*). *James et al., 2016* pointed out some limitations of TE in detecting information flow, they showed through simple examples of how TE can overestimate flow and underestimate the influence. On the other side, *Garofalo et al., 2009*, *Ito et al., 2011* and *Orlandi et al., 2014* showed how using a combination of time delays and adequate binning may overcome these limitations and make TE analysis a powerful tool for inferring effective connections. Furthermore, *Nigam et al., 2016* discussed the importance of handling firing rate and population bursts influence, as well as spurious connections caused by common drive and transitivity to have a more accurate effective connections inference.

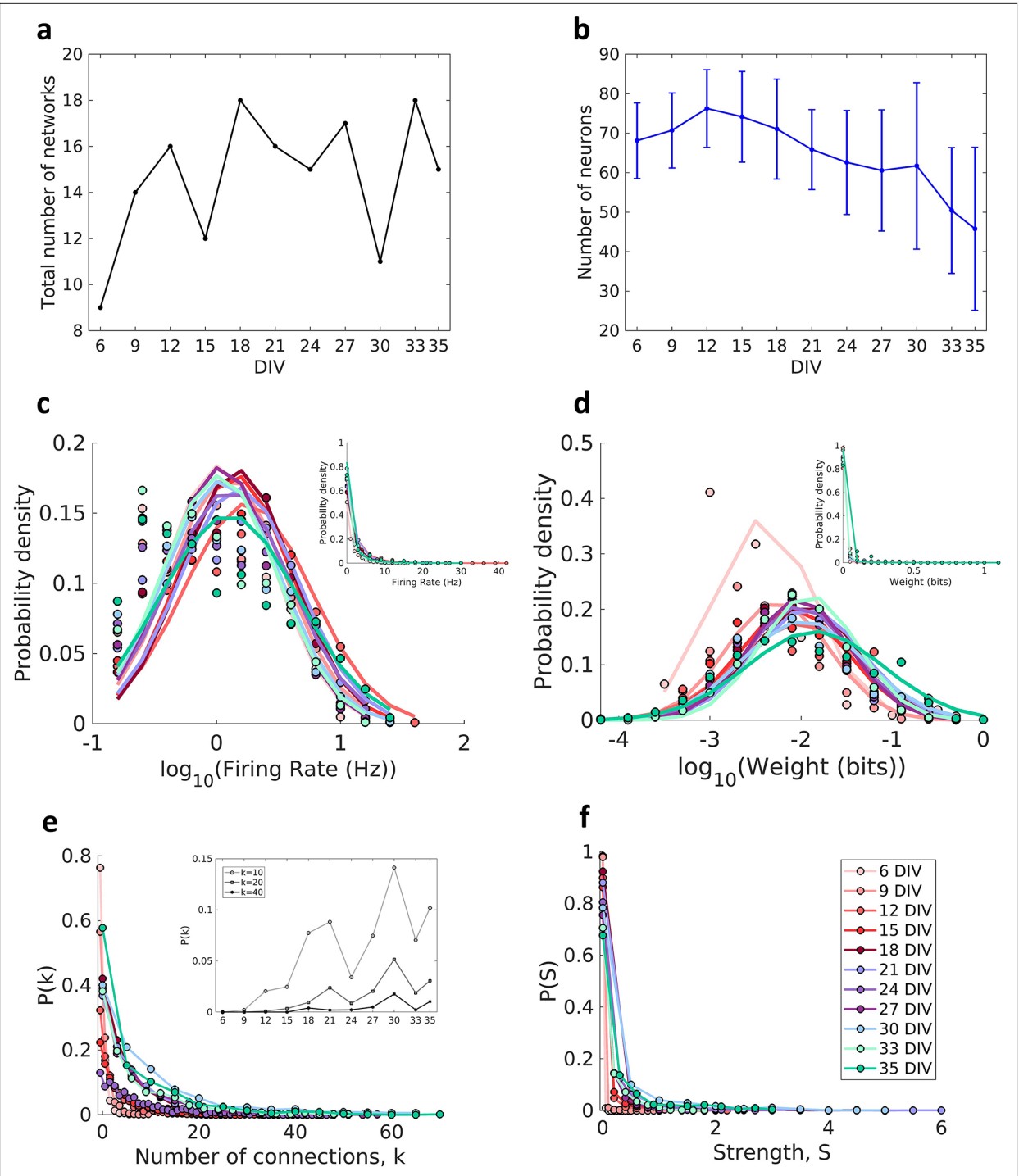

**Figure 1.** Description of the networks. Neuronal cultures were recorded from 6 to 35 days in vitro (DIV). (**a**) Number of the total networks analyzed for each considered DIV. (**b**) Number of neurons recorded in the cultures for each DIV (mean ± SEM). Sample size for each DIV: 9, 14, 16, 12, 18, 16, 15, 17, 11, 18, and 15. (**c**) Distributions of the pooled logarithm firing rate of neurons recorded in each DIV represented by the Gaussian fit of the computed values. Neurons with a firing rate lower than 0.2 Hz were cut off. Inset: distributions of the pooled firing rate of neurons recorded in each DIV fitted by the generalized Pareto distribution. (**d**) Distributions of the pooled logarithm weights represented by the Gaussian fit of the normalized transfer entropy resulting from the significant connections (see Materials and methods). Inset: distributions of the pooled weight values computed in each DIV fitted by the generalized Pareto distribution. (**e**) Distributions of the pooled number of connections per neuron by DIV. Inset: probability of connections with k = 10, 20, and 40 for each DIV. (**f**) Distributions of the pooled strength were defined as the sum of the weights of the connections per neuron by DIV.

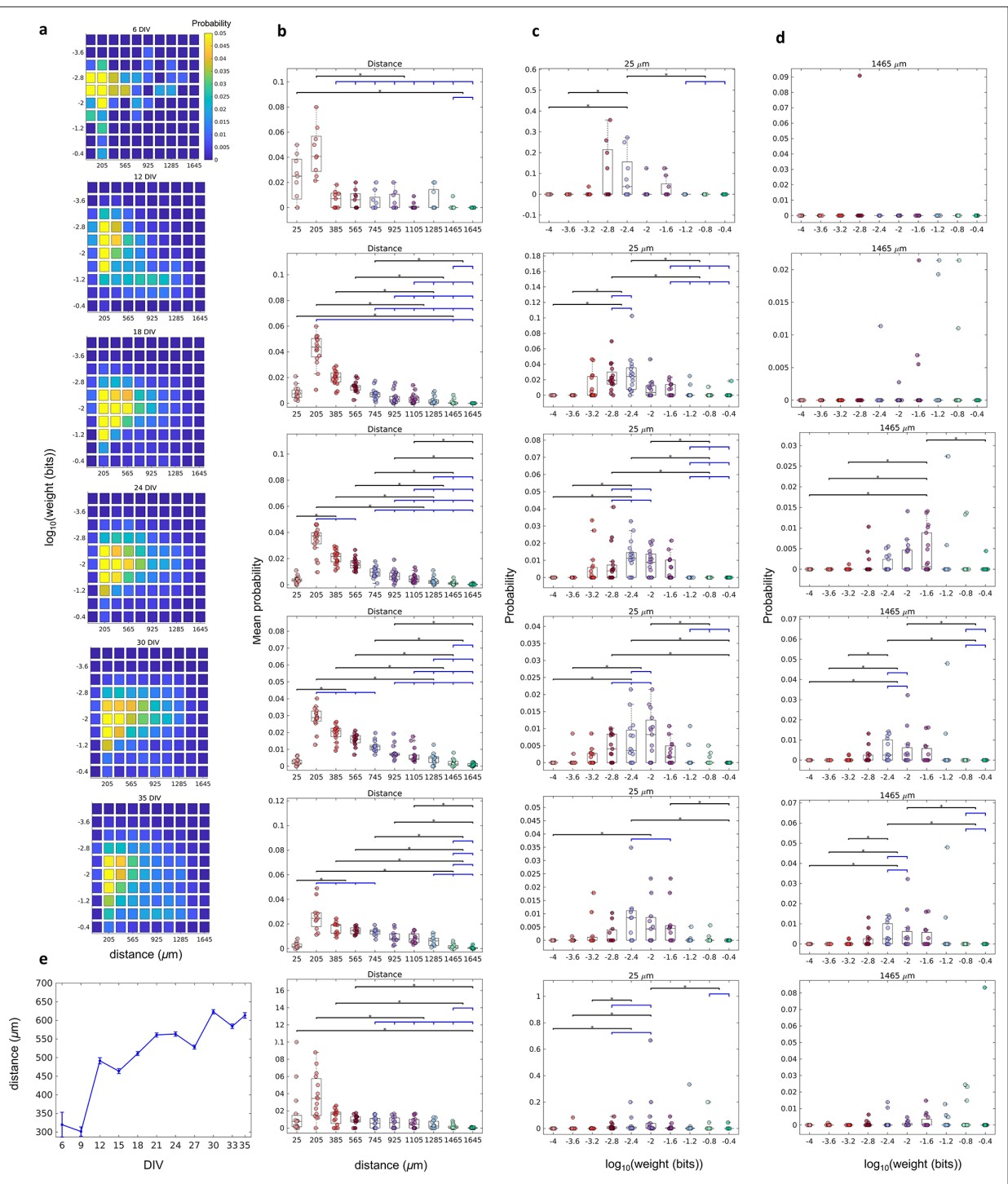

**Figure 2.** The relationship between the weight of effective connection and the distance between neurons. (**a**) Joint probability distributions of the pooled logarithm weight and the distance between neurons. Weights were defined as the normalized transfer entropy resulting from significant connections while distance was calculated as the Euclidean distance between the electrodes from where the two neurons involved in a connection were recorded. (**b**) Mean probability density by distance. (**b–d**) Each row relates to 6, 12, 18, 24, 30, and 35 days in vitro (DIV), respectively. (**c**) Probability density only for 25 μm of the distance between neurons for each connection weight (logarithm). (**d**) Probability density only for 1425 μm of the distance between neurons for each connection weight (logarithm). Statistical tests: Kruskal-Wallis test followed by the Tukey-Kramer post-hoc test. (**e**) Distance between neurons over DIV (mean ± SEM). Sample size for each DIV: 4, 13, 16, 12, 18, 16, 15, 17, 11, 18, and 11. Asterisks indicate p-values ≤ 0.05 (*).

The online version of this article includes the following figure supplement(s) for figure 2:

**Figure supplement 1.** The relationship between the weight of effective connection and the distance between neurons.

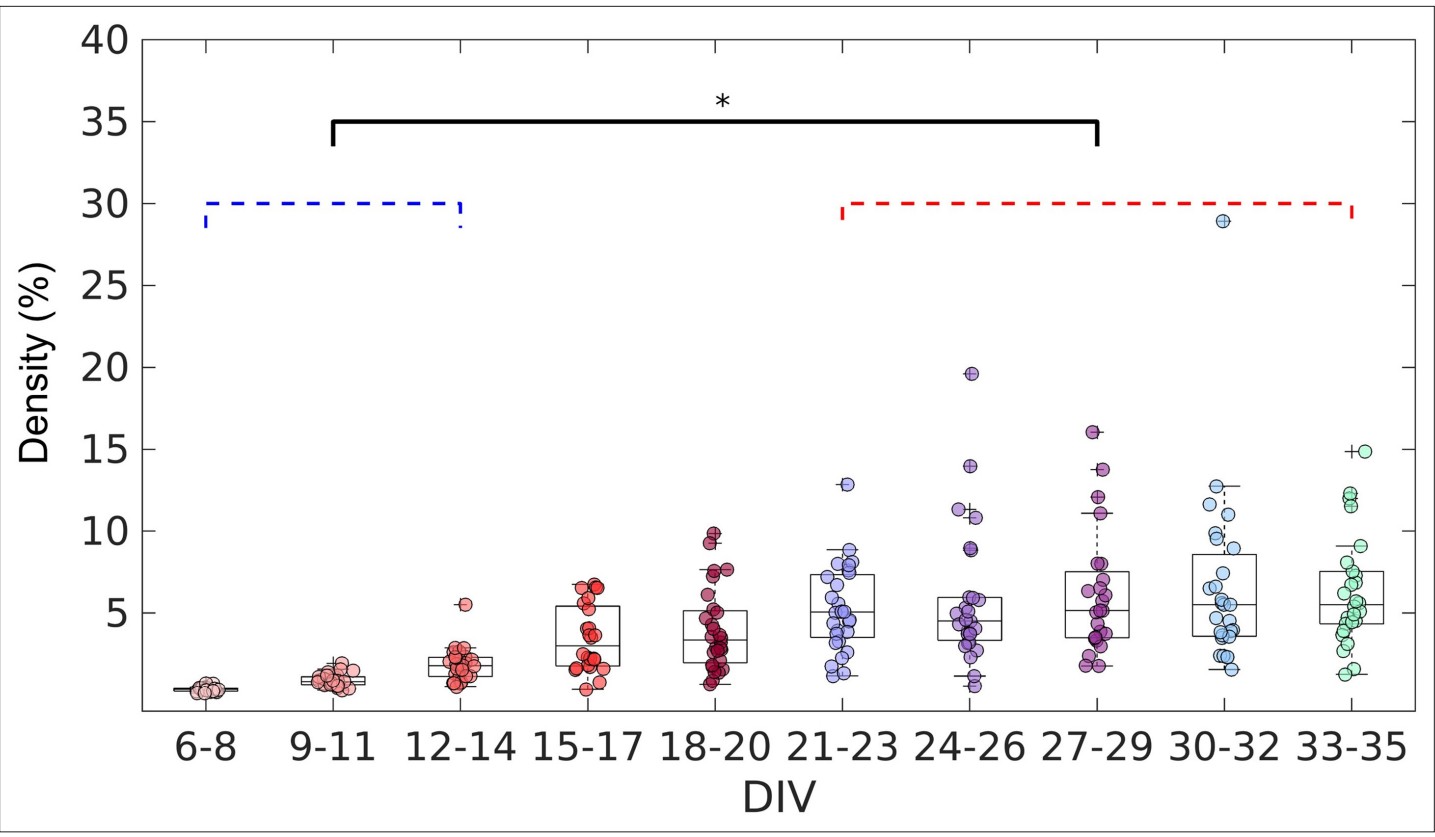

**Figure 3.** Evolution of the edge density of effective networks over time. Edge density was computed by normalizing the number of actual connections by the total number of possible connections (N²–N, where N is the total number of nodes). The edge density per period was defined as the median for each culture (28 cultures) among 3 days. Statistical tests: Kruskal-Wallis test followed by the Tukey-Kramer post-hoc test. Asterisk indicates p-value ≤ 0.05 (*).

Underpinned by these works we constructed the present methodology. To test the accuracy of the effective connectivity inference we used Izhikevich's network model (*Izhikevich, 2006*). The network dynamics was constructed based on the synaptic weights and delays between neuron activity propagation. The model reflects information flow pathways based on structural connections between cortical excitatory and inhibitory neurons. The mean firing rate of excitatory neurons was 4.78 ± 0.84 Hz (mean ± SD) and of inhibitory neurons was 16.82 ± 2.19 Hz (mean ± SD). This model helps us to test the ability of the pipeline in detecting the predictive information transfer. Since we compared the transferred information computation with the structural couplings of neurons that generated the information transfer.

To quantify the ability of the methodology adopted in this work to predict the known connections in the modeled networks, we computed the area under the receiver operating characteristic (ROC) curve. This metric provides an objective and non-parametric way to measure how the algorithm classified the connections, taking into account the number of true positive, true negative, false positive, and false negative cases in the connection inference. A result equal to 1 indicates that the algorithm was able to detect where there was and where there was no synaptic connection in the model with 100% of precision. We found an area under the ROC curve equal to 0.8548 ± 0.0037 (mean ± SD).

### Topological measures

After confirming the effectiveness of the TE analysis to infer the connections, we analyzed the topology of the effective networks over 29 DIV. We found an increase in the edge density over time (*Figure 3*), as also observed by *Downes, 2012* and *Schroeter et al., 2015*. The edge density increased for the first 21 DIV (red bracket in *Figure 3*) before the networks stabilized at a density of 5% ± 1.9% (median ± interquartile range, IQR).

*Figure 4a* shows that in the first DIV, the networks were fragmented into disconnected components and as the culture matured, a single giant component emerged (*Newman, 2003*, number of components significantly equal to 1 on the 18–24, 30 DIV, p<0.05). To correct for the possibility of bias, we used only the largest connected components in our analysis. If no component had more than five neurons, the culture was discarded.

## Segregation and integration

To better characterize the architecture of the effective networks, we measured the clustering coefficient, average path lengths, and small-worldness (see *Figure 4b–d*). Initially, on the 6 and 9 DIV, the clustering coefficient was not significantly different from random networks. As the edge density increased, the clustering coefficient increased, while the path length was the same as in random networks for most of the DIV (p=0.0464, p=0.0250, and p=0.0147 for 21, 24, and 30 DIV, respectively). After 12 DIV, the networks started to have a clustering coefficient higher than in random networks (p<0.05), which implied a small-world organization of the effective networks after the same period (p<0.05). The increase in the clustering coefficient resulted in the emergence of a segregated architecture, while the integration was maintained by the shortest path length, as shown in *Figure 4b and c*.

When the 13 unique three-node structural motifs (as described by *Sporns and Kötter, 2004*) are considered, only 5 of the 13 patterns (5, 8, 11, 12, and 13) were significantly higher (p<0.05) than in random networks for the DIV shown in *Figure 4e*. The total frequency of motif occurrence in the whole network (fingerprint) was computed for each pattern found in the actual network normalized by the same coefficient for the correspondent random network. The frequency fingerprint of motifs 5, 11, and 12 was higher than in random networks after 12 DIV. Whereas motifs 8 and 13 started to present a frequency fingerprint higher than in random networks later in development, after 15 DIV, and only within 21 and 27 DIV, respectively. The results were not deemed significant for motifs 11 and 12 on some DIV, which might be explained by the fall in the fraction of the networks that presented these motifs in the same DIV as shown in *Figure 4e* (results lower than 0.8). The fraction of networks represents the number of networks presenting at least one motif that has presented the pattern. The smaller fraction of networks that presented motif 13 might also be related to the small-time interval in which this pattern is higher than in random networks. According to the fraction of the networks in which each motif appeared, motifs 5 and 8 are easily found in the networks over time, since they were presented in more than 80% of the cultures in most of the DIV. While motifs 11 and 12 show a small fluctuation of around 80% of appearance over time. By contrast, motif 13 is less common, appearing in more than 60% of the cultures only in later stages of maturation.

## Modular organization

To verify if neurons self-organize into small, integrated modules, and if this organization was related to the physical location of neurons, we applied a non-overlapping module detection algorithm to the effective networks. The physical location of neurons within modules for one culture over DIV is shown in *Figure 5a*. The number of modules shown in *Figure 5b* rise until the 12 DIV and then stabilized until the 33 DIV with an average of approximately five modules. Finally, following 33 DIV, the number of modules decreased until 35 DIV. The number of neurons per module followed a Poisson distribution for all DIV (see *Figure 5c*). The higher probability of modules with a smaller number of neurons suggests a preference for smaller-sized network modules. To investigate whether smaller modules were more efficient, which would justify the cause of this phenomenon, the module global efficiency was computed. Indeed, *Figure 5d* shows a negative Spearman's correlation between the module global efficiency and the total number of neurons within the module for all DIV (rho ≥–0.2, p<0.05). In other words, modules with a smaller number of neurons presented a higher global efficiency. Additionally, the physical distance between neurons within modules was inferred based on the position of the electrodes that recorded a given neuron's activity. *Figure 5e* shows the positive Spearman's correlation between these two measures after 9 DIV (rho ≥0.22, p<0.05), indicating that neurons within smaller modules were closer together.

Previous studies have proposed that a modular network topology optimizes the trade-off between wiring cost minimization and functionality efficiency in the brain (*Betzel and Bassett, 2017*; *Meunier et al., 2010*). To investigate how the division of the effective networks into modules relates to wiring

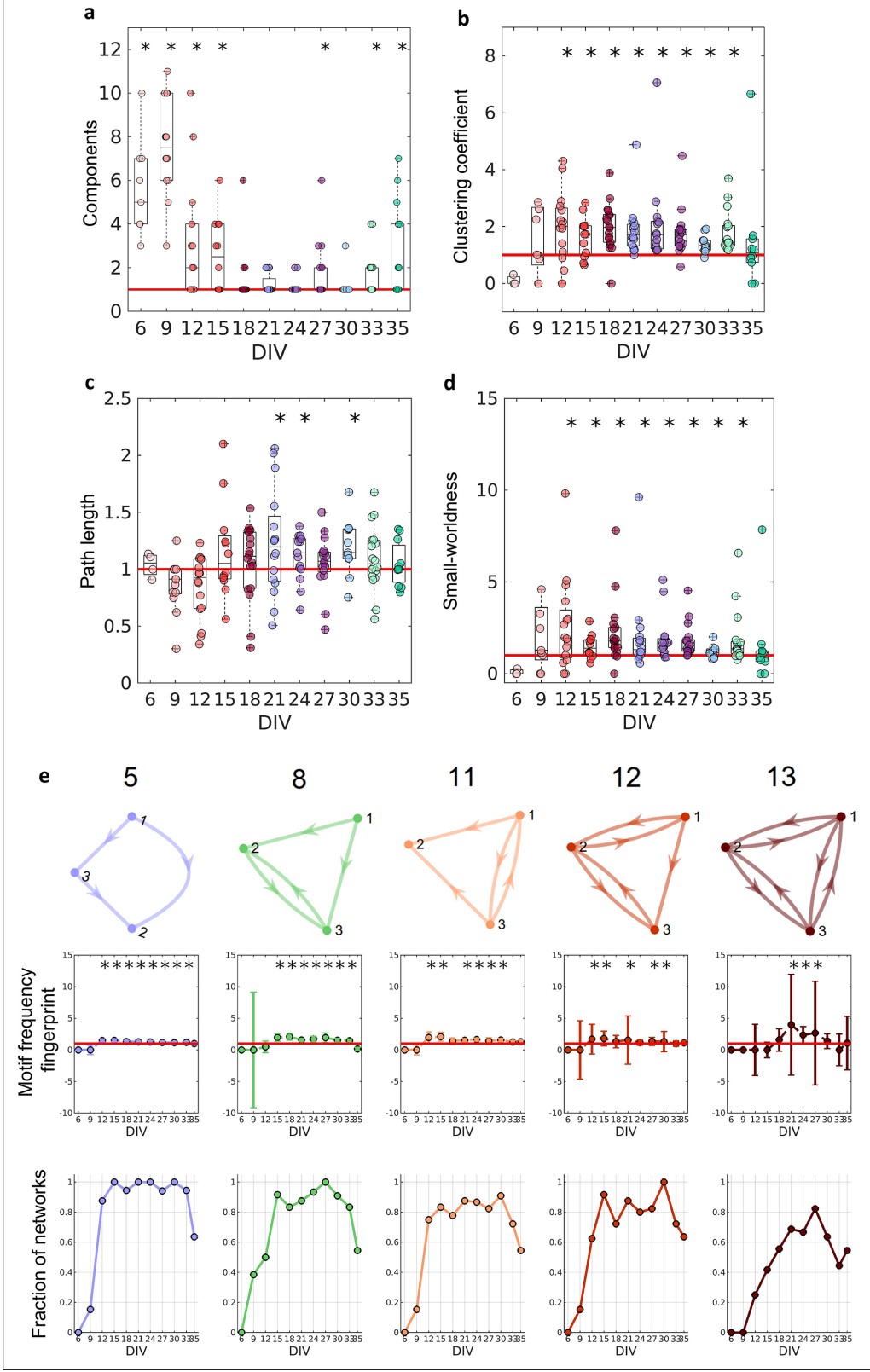

**Figure 4.** Graph-theoretic measures over maturation. Various network measures were computed from effective networks in different periods of maturation. (**a**) Number of components computed for each network by days in vitro (DIV). The red line represents the convergence of the entire network to only one large and fully connected component. (**b**, **c**) Global clustering coefficient, and path length, respectively. The results for both coefficients

*Figure 4 continued on next page*

*Figure 4 continued*

in actual networks were normalized by the average result from 100 randomized networks (see Null model in the Materials and methods section). The red line represents when the normalization is equal to 1 meaning the results for actual and random networks are the same. (**d**) Small-worldness results considering the actual clustering coefficient and path length compared to a random network. The red line represents that the network architecture is not a small-world. (**e**) Motifs that were found significantly more frequent than in random networks during maturation. The motif frequency fingerprint was considered as the total motif frequency of occurrence in the whole network; results were normalized by the same measures in random networks (values are presented as mean ± SEM). The red line represents where the frequency of the motif is the same as in random networks. The fraction of networks was computed by dividing the number of networks that presented the motif by the total number of networks that presented at least one motif. Sample size for each DIV: 4, 13, 16, 12, 18, 16, 15, 17, 11, 18, and 11. To test whether the results come from a population with a median equal to 1 the Wilcoxon signed-rank test was used. Asterisks indicate p-values ≤ 0.05 (*).

cost, we compared the sum of the physical connection lengths inside and outside modules, normalized by the sum of total physical connection lengths for actual and random networks over DIV. This comparison showed that after 12 DIV the wiring cost inside modules was significantly higher (p<0.05) in actual networks than in random ones (*Figure 5f*), until 30 DIV. Conversely, the wiring cost outside modules was lower (p<0.05) for actual networks than for random ones after 9 DIV and until 27 DIV.

## Neuronal hubs

The similarity of the degree distribution with heavy-tailed distribution for all DIV indicates the existence of a few highly connected neurons (hubs). Hubs bridge different communities, facilitating the integration of information through the network. Many works have defined hubness as a function of node degree. However, in this work, we have a narrower and more conservative definition. Based on *Sporns et al., 2007*, we considered not only the number of connections of a node but also its strength (the sum of connection weights), as well as centrality indices (betweenness centrality and closeness centrality). This analysis captures not only highly connected nodes but also nodes with important connections that lie on many of the shortest path lengths of the network. To investigate the emergence and role of these important nodes during the maturation of the networks we ranked the results for each neuron in the same culture, considering all DIV in which the culture was recorded, taking into account the degree, strength, betweenness centrality, and closeness centrality. Considering the 40% of the highest values for these four coefficients, neurons that ranked in all of them were classified as having a score of 4. If they ranked only for three coefficients (independently of which one), they were classified as having a score of 3, and so on for 2 and 1. Through this ranking approach, we were able to establish a definition of node importance in the network topology to help us understand its role in the connections between modules.

*Figure 6a* illustrates how neurons with different scores were physically distributed by one culture after 21 DIV. Considering the fraction of neurons for each score over DIV, *Figure 6b* shows that neurons with a score of 4 emerged only after 9 DIV. Between 21 DIV and 27 DIV, there was no difference among the fraction of neurons of any score. The results around 0.2 for each score at this period imply that ~0.2 of neurons did not present any score, indicating a similar division of neurons among these 5 groups.

The weighted rich-club coefficient definition was adapted to be computed as a function of scores (see Methods for additional details). This allowed us to investigate how the fraction of the highest weighted connections outside modules was distributed by neurons with different scores (*Figure 6c*). *Figure 6d* shows an increase in the rich-club coefficient as the score number increases. This result indicates a tendency for stronger connections between neurons with higher scores, more than 'by chance,' outside modules. The values for neurons with a score of 4 and 1 were deemed significant (p<0.05) for most of the DIV, indicating the difference between these two groups of neurons.

Furthermore, we computed the participation coefficient ($P_c$) for neurons with different scores. Similarly to the results for the rich-club coefficient, the median $P_c$ results were higher for neurons with higher scores (*Figure 6f*). Neurons with a score of 4 and 1 presented results deemed significant (p<0.05) since 9 DIV. It also suggests a difference between them. Using the definition of $P_c$, given by *Guimerà and Nunes Amaral, 2005*, neurons with a score of 4 seem to have a connector role, once they have many connections with most of the other modules (median $P_c$>0.3). While neurons with a

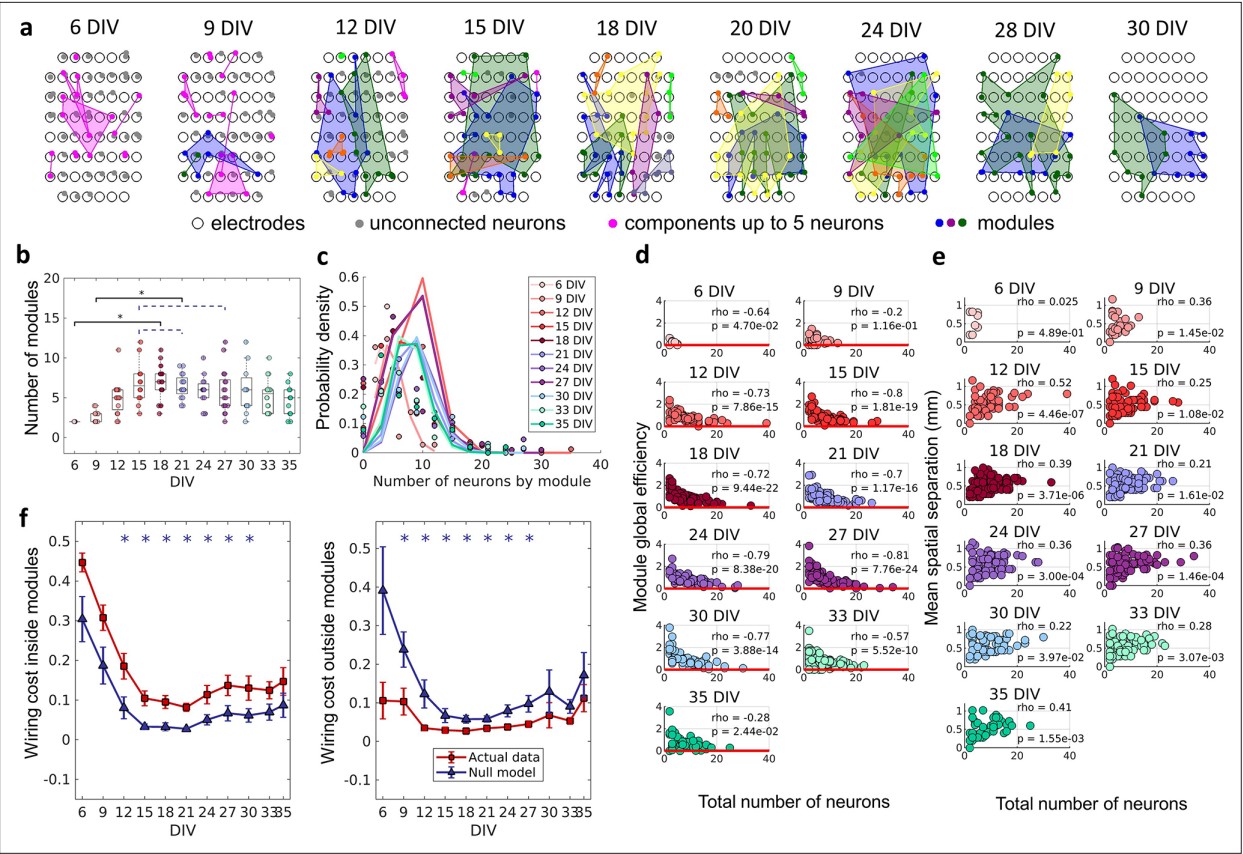

**Figure 5.** The emergence of modular organization. A community detection algorithm allowed to track the division of the networks into non-overlapping modules. (**a**) Schematic representation of the physical location of modules in culture over days in vitro (DIV). Connected and colored areas link the neurons within the same modules. (**b**) Mean number of modules for each network by DIV. Statistical tests: Kruskal-Wallis test followed by the Tukey-Kramer post-hoc test. (**c**) Distributions of the pooled number of neurons by module represented by the Poisson fit of the computed values by DIV. (**d**) Global efficiency was computed for each module and normalized by the averaged coefficient considering the same neurons in 100 random networks (see Null model in the Materials and methods section). The values shown are the pooled normalized coefficients for all the modules in all cultures for the same DIV. The red line represents where the module's global efficiency is the same as in random networks. The correlation between the module's global efficiency and the total number of neurons per module was calculated by using Spearman's rho. (**e**) Spatial separation was computed by averaging the length of all connections within each module. The connection length was calculated by the Euclidean distance between the electrodes that gathered the signal from the two neurons involved in each connection. The values shown are for all modules in each culture per DIV. The correlation between the spatial separation and the total number of neurons per module was calculated by using Spearman's rho. (**f**) Wiring cost computation was performed by summing up the Euclidean distance of the electrodes that gathered the signals from the two neurons related to each connection. Results for actual networks are presented as mean ± SEM considering the connection inside and outside modules normalized by the total wiring cost of the network separately. Additionally, results for the null model were averaged from 100 random networks, taking the same neurons for each module but different connections. Sample size for each DIV: 4, 13, 16, 12, 18, 16, 15, 17, 11, 18, and 11. Actual and modeled results were compared on the same DIV by using the ANOVA. Asterisks indicate p-values ≤ 0.05 (*).

score of 1 have a more peripheral role, having more connections within their module (median $P_c < 0.3$). Whereas neurons with a score of 3 and 2 are in the middle.

Regarding specific characteristics that could influence neuron score, we found evidence of a relationship between firing rate and score. **Figure 6e** shows that the higher the neuron score, the higher the median firing rate over all DIV. In general, after 12 DIV, the firing rate of neurons with a score of 4 and 3 was significantly higher than neurons with a score of 1 (p<0.05), and in some periods, (18, 21, 27, 30, and 33 DIV) the firing rate of neurons with a score of 4 was higher than neurons with a score of 2. Collectively, these findings suggest the behavior and dynamics of individual neurons, and neuronal assemblies, drive the emergence of complex effective network topologies. Overall, a small-world architecture composed of modules with most likely few and nearby neurons, which are integrated by more topologically important neurons with a high firing rate.

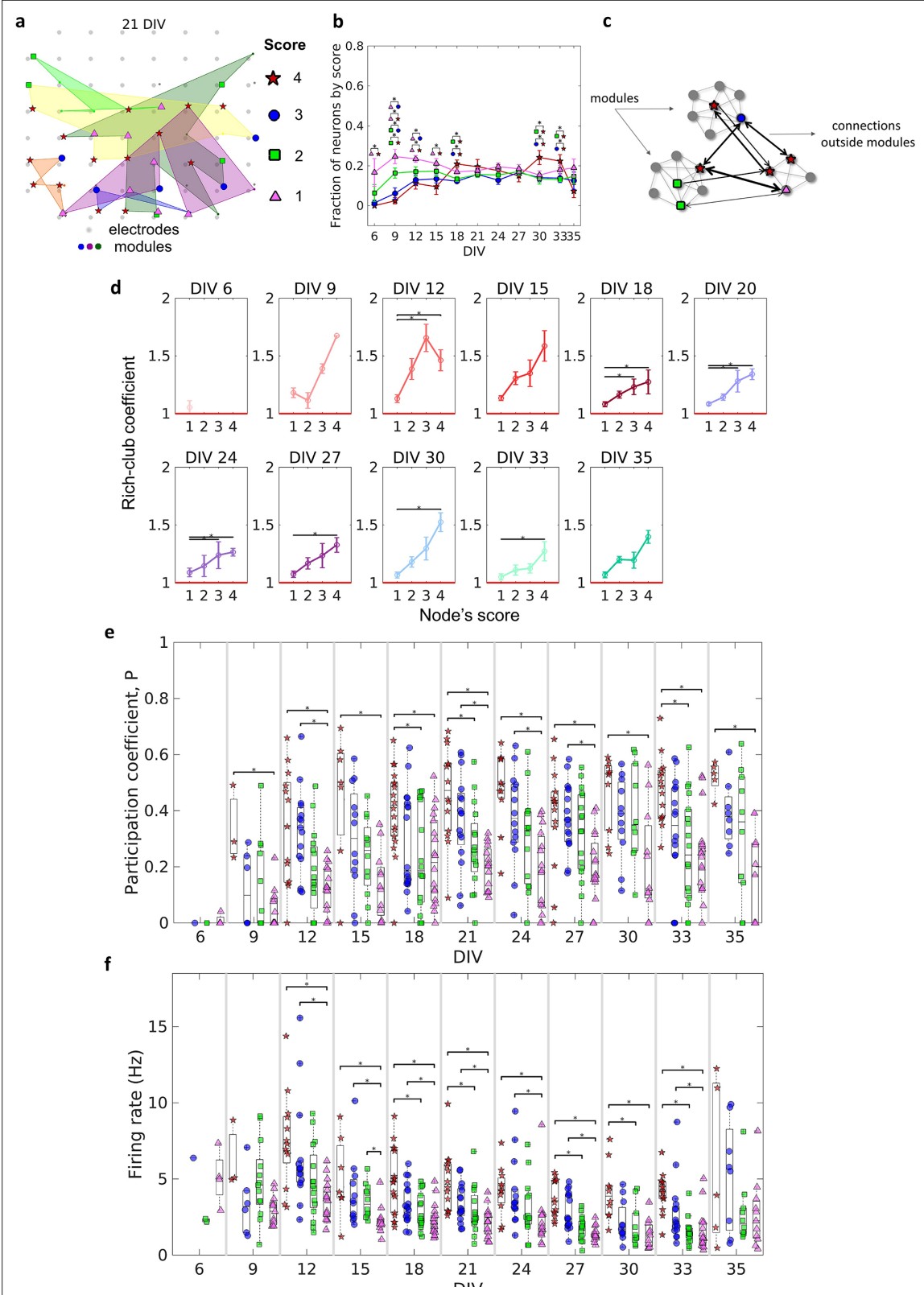

**Figure 6.** Topologically important nodes and their role as module integrators. Neuron hubness was classified by scores. Four measures were computed for each neuron, degree, strength, betweenness centrality, and closeness centrality. The score of a neuron was based on the number of measures that it ranked in the highest values.(**a**) Graphical representation of the location of nodes with different scores in one culture after 21 days in vitro (DIV). (**b**) Evolution of the fraction of neurons by score, defined as the number of neurons for each score normalized by the total number of neurons in the

*Figure 6 continued on next page*

*Figure 6 continued*

culture, over DIV. Results are presented as mean ± SEM, they were compared on the same DIV by using the one-way ANOVA. (**c**) Graph representation of outside modules connections used in the weighted rich-club coefficient computation. Black arrows represent the connections used. (**d**) Weighed rich-club coefficient computed considering only the connections highlighted in **c**. Values are presented as the mean ± SEM for actual networks normalized by the average value for 100 random networks, where the same neurons but different connections were considered. The red line represents where the rich-club coefficient is the same as in random networks. (**e**) Mean participation coefficient for neurons with each score in every culture over DIV. (**f**) Mean firing rate for neurons with each score in every culture over DIV. Sample size for each DIV by score: 6 DIV {**4**: 0; **3**: 1; **2**: 2; **1**: 4}; 9 DIV {**4**: 3; **3**: 6; **2**: 13; **1**: 13}; 12 DIV {**4**: 13; **3**: 15; **2**: 16; **1**: 16}; 15 DIV {**4**: 7; **3**: 12; **2**: 12; **1**: 12}; 18 DIV {**4**: 18; **3**: 18; **2**: 18; **1**: 18}; 21 DIV {**4**: 14; **3**: 16; **2**: 16; **1**: 16}; 24 DIV {**4**: 11; **3**: 14; **2**: 15; **1**: 15}; 27 DIV {**4**: 14; **3**: 17; **2**: 17; **1**: 16}; 30 DIV {**4**: 10; **3**: 11; **2**: 10; **1**: 11}; 33 DIV {**4**: 15; **3**: 16; **2**: 18; **1**: 18}; 35 DIV {**4**: 5; **3**: 8; **2**: 10; **1**: 11}. Statistical tests: Kruskal-Wallis test followed by the Tukey-Kramer post-hoc test. Asterisks indicate p-values ≤ 0.05 (*).

## Discussion

Here, we present an analysis of how the topological properties of effective networks change during the development of in vitro neuronal assemblies. Effective networks showed non-random topological properties over time associated with the ongoing activity in individual neurons. Our results suggest that both the physical distances between neurons, as well as the heterogeneity of firing rates are key factors in the development of effective network organization. As the complex network topology emerged, we were able to investigate how it relates to the self-organization of neuronal assemblies.

## Changes over time

We found that most of the metrics reported here underwent significant changes after 12 DIV. *Figure 3* shows an increase in the edge density during the 12–14 DIV period. Additionally, *Figure 4a* also shows that some networks had already started to present only one component after 12 DIV, indicating fully connected networks. Prior works investigating the dynamics of electrophysiological activity of dissociated hippocampal neuronal cultures maturation also found an increased number of network bursts at the same period (12–14 DIV) than in previous periods of the maturation (*Biffi et al., 2012*; *Cohen et al., 2008*). Network bursts indicate the synaptic spreading of the action potential across neurons reflecting the effects of a fully connected network (*Hales et al., 2010*). A significant number of edges and the integration of the whole network might have been responsible for the emergence of complex network framework properties.

Considering functional networks *Downes, 2012* argued that the small-world topology emerges from a random network. In contrast, our findings suggest that neurons are promoting neurite outgrowth and refining the effective connections to form a small-world architecture (or other complex network properties) since they are plated. During the early phases of development, the topology of the neuronal effective network is resembling a random network. However, this does not necessarily mean the connections are randomly established, but it may indicate that neurons do not have enough connections to form a complex framework. In this way, the complex architecture might emerge by inserting new connections and not by changing the existing topology. This hypothesis is supported by most of the results stabilizing over the 12–30 DIV period when the edge density increased to around 5% and the number of components decreased to around 1. Even though after 12 DIV the networks started to present complex network organization, the edge density kept growing and achieved stabilization only after 21 DIV. These results show that development is a multi-stage process that takes place over the course of weeks.

Conversely, after 30 DIV, the networks had a change in topology, ultimately returning to an architecture similar to a random network. As we used recordings from neurons in culture, this effect might be explained by neuronal death associated with the difficulties of maintaining a healthy cell culture beyond 4 weeks (*Kaech and Banker, 2006*). Indeed, *Figure 1b* shows a decrease in the number of recorded neurons, although no significant decrease in the number of network edges has been detected (*Figure 3*). The impact on the topology after such a period includes a breakup of the networks into different components, as can be seen in *Figure 4a*. This result can also indicate how the indiscriminate loss of neurons may drastically affect neural circuits during pathological conditions. However, the silencing of neurons may be a result of the multi-stage process of circuit refinement during development. To test this hypothesis novel in vivo studies should be performed.

## Neuronal assemblies self-organization

Considering that dissociated neurons in a culture set up firstly structural connections that are refined by activity-dependent factors, patterns of effective connectivity might be a result of combined self-reinforcing mechanisms and a dispute for limited resources. *Zheng et al., 2013* related the self-reinforcing with Hebb's postulate of synaptic plasticity, which establishes that two groups of neurons strengthen their synaptic connections when their firing patterns are correlated. And the competition for limited resources with the neuronal homeostatic mechanisms that scale the neural plasticity, where one synapse can only be strengthened by weakening another, making the sum of all the synapse weights of a neuron slightly constant.

In agreement, the 'silent' synapse concept seems to have an important role in neuronal assembly formation and plasticity (*Kerchner and Nicoll, 2008*). *Liao et al., 1999* studied the presence of glutamate AMPA and NMDA receptors on silent synapses of hippocampal cultures during maturation to investigate their role in the alteration of synaptic strength. NMDA receptors are known for their modulatory mechanisms associating voltage-dependent magnesium blockage and high calcium permeability (*Iacobucci and Popescu, 2017*), whereas AMPA receptors act in fast excitatory transmission (*Derkach et al., 2007*). The authors showed that silent synapses are composed exclusively of NMDA receptors, indicating that presynaptic glutamate release does not generate an excitatory postsynaptic potential in the postsynaptic neuron. Such synapses can be activated by acquiring functional AMPA receptors by NMDA-receptor-dependent long-term potentiation (LTP) mechanisms as previously reported by other work (*Kerchner and Nicoll, 2008*). The authors also showed that during the first week in vitro, almost 95% of the synapses were silent, in the second week around 65%, and at about 40% in the third week, while the number of synapses expressing AMPA receptors increased. Although these results suggest the transition from silent to active synapses, one could question the role of inhibitory neurons in these interactions. *Ben-Ari et al., 1997* reviewed how the principal transmitter of inhibitory signaling in the adult central nervous system, GABA$_A$ receptors may help to activate silent synapses. GABA$_A$ receptors act differently in immature neurons (until the second postnatal week in rats, *Ben-Ari et al., 1997*, 7–8 DIV in cultures, *Soriano et al., 2008*), their activation leads to a cell depolarization which was considered truly responsible for the potentiation of NMDA receptors during the activation of a silent synapse.

When all these are considered, a possible interpretation is that after being plated the hippocampal neurons start to grow processes and make connections firstly with nearby neurons, and as the dendrites and axons outgrow with more distant neurons. However, during this period, the synapses are most likely the silent type. The synapses are activated by the induction of spike-timing-dependent plasticity (STDP) only where there is a time coincident fire between the pre and postsynaptic neurons, and/or with the GABA depolarization aid. In these cases, the activation of a synapse gives rise to an effective connection.

Furthermore, spatial distribution and firing rate of neurons may have an important role in the regulation of the synapse weights, interfering with the network structure, and consequently, the effective network topological properties. Once the interaction with neighboring neurons takes place first if the synapses among them are activated the $[Ca^{2+}]_i$ can reach an optimal point and make neurons stop outgrowing (*Kater and Mills, 1991*). However, while neuron activity is not stabilized the neurites keep outgrowing and seeking for neurons to synchronize. Jointly, considering that the synapses are activated by synchronized spontaneous activity, if the firing rate of one or both neurons is higher it is most likely that pre- and postsynaptic neurons synchronize and consequently activate the synapse between them.

Indeed, the results of connection weight by connection distance (*Figure 2*) showed that most of the edges connect neurons that are nearest in the first DIV, this distance had a slight increase after 12 DIV, and then it is sustained after network maturation. Besides that, *Ito et al., 2014* found a higher density of connection for smaller distances and an exponential decay as the distance increased in organotypic hippocampal cultures. The results presented by *Zheng et al., 2013* also corroborate this hypothesis. The authors showed that STDP and scaling synapse mechanisms play an important role in the formation and maintenance of neural circuits, and they are essential to keep the connection strength distribution of the networks within the log-normal shape. We showed that the connection weights distribution has had a log-normal shape since the first day recorded (*Figure 1d*), suggesting the participation of such mechanisms.

Although this work focused on the effective connectivity during the self-organization of neuronal assemblies, the relationship between structural and functional/effective connectivity has been a subject of interest in many other works (*Meier et al., 2016*; *Segall et al., 2012*; *Suárez et al., 2020*). To relate structural and effective connectivity is not an easy task since inhibitory activities do not allow all possible communication paths to be explored when effective connectivity is inferred from neuronal dynamics. As pointed out by *Park and Friston, 2013* structural networks constrain functional networks but also many patterns of functional connectivity can emerge from fixed structural ones and give rise to high-level neurocognitive functions. In this way, the silent synapses hypothesis might also explain functional connectivity results. However, the use of spatially organized electrical stimuli through the electrodes could support a better way to establish a relationship between neuronal structure and function by evoked activity (*Bauer et al., 2018*).

## Emergence of complex networks properties from self-organization

The total square recording area formed by the electrodes corresponds to about 1.5 × 1.5 mm². The lateral size of the square corresponds to the axonal growth of neurons in culture (*Kaneko and Sankai, 2014*), which, in principle, could foster a neuron to connect with any other in the culture. However, when we compared the topological measures for inferred networks with random networks the results were significantly different, showing the emergence of non-random topological properties and neurons' tendency to connect with their neighbors.

Complex networks are defined as having highly interconnected units forming an irregular framework that evolves dynamically over time (*Boccaletti et al., 2006*). The study of the topology presented in this kind of network is an active area of research. The most-reported properties in empirical studies are the node degree, that is, the number of directed connections of a node, deviating from a Poisson distribution as presented in random networks to a power-law (scale-free, *Strogatz, 2001*), a short path between any two nodes (*Amaral and Ottino, 2004*), and the presence of specific short-cycle patterns or motifs (*Boccaletti et al., 2006*). Regarding networks inferred from neuronal populations, different works have already shown that neurons cluster together during the network development to form a complex architecture, both structurally (*de Santos-Sierra et al., 2014*; *Shefi et al., 2002*; *Teller et al., 2014*) and functionally (*Downes, 2012*; *Schroeter et al., 2015*), but it remains an open question how this organization is guided. Effective connectivity indicates how the activity of one neuron directly influences the activity of another neuron. By looking at how effective network topology develops during the maturation of neuronal assemblies, we may infer how specific architectures emerge from neuronal interactions.

Small-worldness is a common complex network characteristic found in brain networks (*Bassett and Bullmore, 2006*). In our results, the emergence of this architecture is largely due to the clustering coefficient, given the path length did not significantly change for most of the DIV. The small-world property reflects a network configuration able to maximize communication efficiency while minimizing cost. The emergence of this architecture has been modeled in different ways. *Watts and Strogatz, 1998* proposed a random rewiring scenario in which a few edges of a lattice network were randomly rewired, creating shortcuts and a topology that combined high clustering and short path length. Moving to a neural perspective, *Kwok et al., 2007* modeled a Hindmarsh–Rose spiking neuron network with random connection patterns rewired as a result of neuron synchrony. Briefly, they compared the synchronization of spikes (or bursts) of a chosen neuron with the entire network and kept or created a connection only for highly synchronized neurons and disconnected neurons with low synchrony. Interestingly, their model presented a substantial increase in the clustering coefficient and a small increase in the path length, leading the networks to a small-world architecture after some steps of rewiring. However, they discussed that this scenario of synchronicity comparison among unconnected neurons is possibly unrealistic. Reasonably, unconnected neurons could not 'sense' their degree of synchronization, given that as far as is known the substrate of neuronal communication is the synapse. Besides that, an extensive rewiring of synapses is extremely unlikely once such a process would require a high amount of time, wiring cost, and energy (*Zhang and Poo, 2001*). In this way, we believe that the activation of silent synapses by the synchronicity between neurons is a better explanation for the small-world emergence. Because it is a more efficient and realistic scenario, supported by empirical evidence.

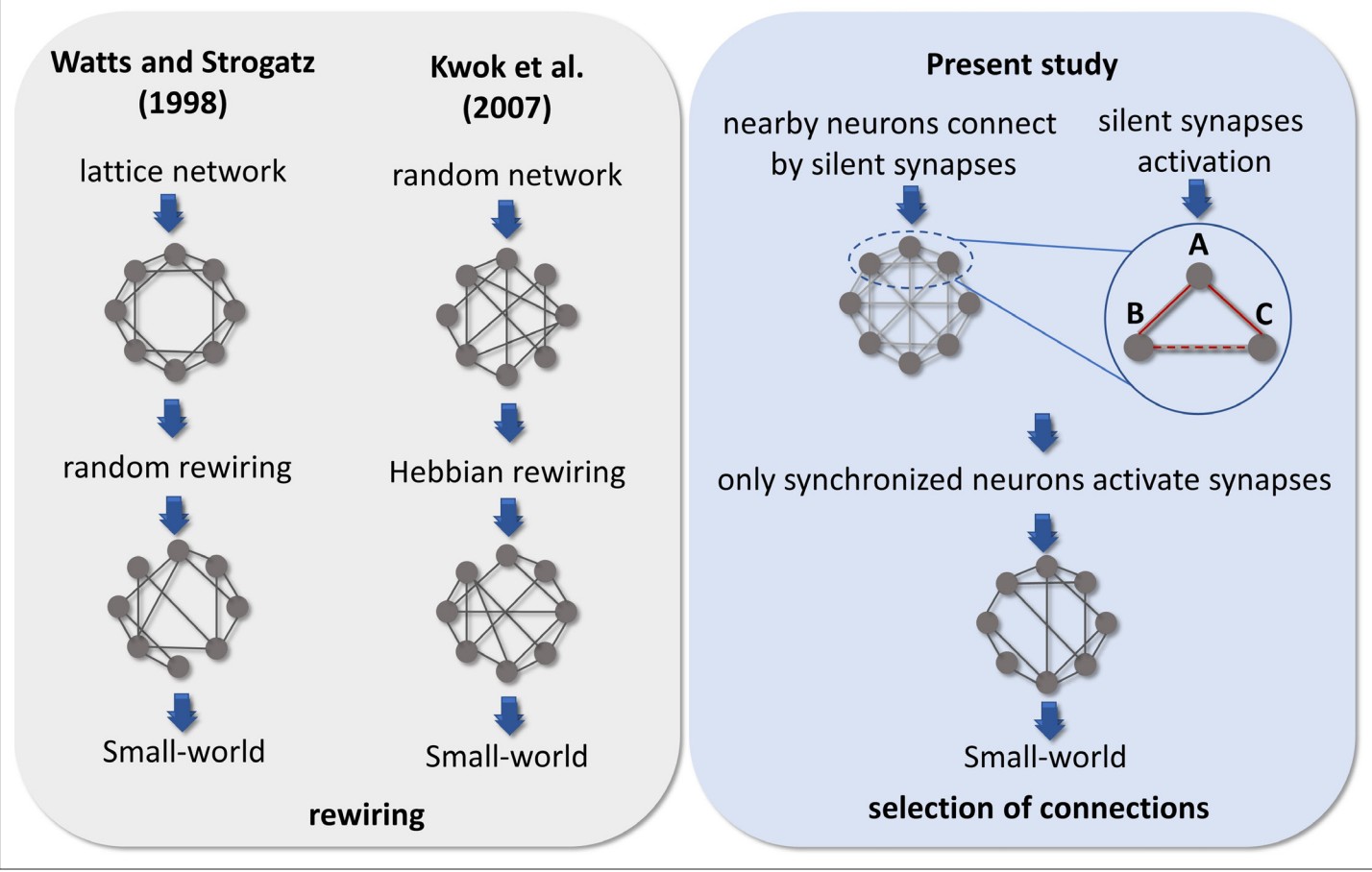

**Figure 7.** Comparison of hypotheses on the emergence of Small-world architecture. Alternative scenarios that might support the development of networks into highly clustered nodes reached by a small number of steps. *Watts and Strogatz, 1998* firstly suggested that a random rewiring of a few edges in a lattice network can lead it to a small-world organization. *Kwok et al., 2007* adapted this hypothesis to the context of neuroscience, suggesting that neurons are initially randomly connected, and these connections are rewired by comparing the synchronization of one neuron to the entire network. Following a Hebbian rewiring rule, highly synchronized neurons receive or maintain a connection while unsynchronized neurons lose it, also leading the network to a small-world organization after a few steps of rewiring. Both hypotheses were raised in a computational model environment, however, the rewiring scenario would involve a high amount of time, wiring cost, and energy to take place in the brain. Considering the results from empirical data reported herein and the silent synapses activation evidence, we hypothesized that neurons might initially make silent synapses with nearby neurons and only synapses between synchronized neurons would be activated. Thus, considering 3 neurons A, B, and C connected by silent synapses. If the synapse between neurons A and B is activated, because they have a synchronized firing activity and the same occurs for neurons A and C. Likely, neurons B and C would also have a synchronized firing activity turning on the synapse between them. The activation of a significant number of synapses gives rise to a Small-world topology in effective networks by a mechanism of selection of structural connections instead of a rewiring process.

The comparison among these hypotheses is summarized in *Figure 7*. Therefore, if we consider three neurons A, B, and C connected by a silent synapse, and that neurons A and B have a synchronized activity that turns on the synapse between them. If neurons A and C also have their synapses activated by neuron synchronicity, likely, neurons C and B would also have a synchronized activity and so they would also have the silent synapse activated. This mechanism is a possible explanation for the increase in clustering coefficient as edge density increases. Conversely, the path length might assume similar values as in random networks for all DIV given the random nature of dendritic arborization and connection with neighboring neurons. Given that NMDA receptors are glutamatergic and voltage-dependent, the temporal lag between pre- and postsynaptic activity is important to the synapse activation (*Durand et al., 1996*). However, the heavy-tailed distributions from which the firing rate log-normal distributions have originated, indicate that most of the neurons have a low firing rate (*Buzsáki and Mizuseki, 2014*). This suggests that the spontaneous synchronization of neuron firings

can be difficult and the GABA$_A$ receptor's excitatory effect may play an important role to tune the network in the first stages of development.

Additionally, the direction of the connections between neighboring neurons is important to determine the network dynamics. Once the local patterns of connection are one of the most important features of neural circuits, studying network motifs may be an interesting way to relate these patterns to how neurons as oscillators are coupled to generate dynamics (*Makarov et al., 2005*).

Using our connection inference methodology, it was not possible to identify which neurons were excitatory or inhibitory. Therefore, it is not possible to discuss the cause of motif formation. However, we can make some general observations. Motif 5 had the most stable results, appearing in more than 85% of all networks after 12 DIV. This motif is the only pattern found that does not involve reciprocal connections, meaning it is a feedforward loop that may support the activity propagation in only one direction (*Beggs and Plenz, 2003*). Motifs 8 and 11 have one reciprocal connection and had a higher variation on the fraction of networks that were present over DIV, most likely because of the higher complexity of a reciprocal connection. As shown by *D'Huys et al., 2008* oscillators coupled bidirectionally can have an attractive or repulsive effect. The same can be noticed for motif 12, with two reciprocal connections and for motif 13, with three reciprocal connections that presented a higher incidence than in random networks only after 21 DIV: the period when the networks stabilized their edge density and integrated into only one component. The combination of all these motifs (5, 8, 11, 12, and 13), comprising excitatory and inhibitory neurons, when organized in a small-world architecture might form the complex framework that gives rise to the zero-lag synchrony during population bursts in mature cultures, as shown in *Penn et al., 2016*. Given that the neuron clustering in these patterns may combine the competition and cooperation of coupled oscillators necessary to synchronize the entire network.

Another widely found complex network property is the subdivision of the networks into modules (*Betzel and Bassett, 2017*; *Boccaletti et al., 2006*). Our results showed that modules most likely having a small number of neurons emerged from the effective networks (*Figure 5c*). Such preference for smaller modules could be explained by the higher global efficiency within modules comprising fewer neurons. The global efficiency relies on the inverse of path length, which itself is the inverse of connection weights. Additionally, the size of the module presented a positive correlation with its spatial dispersion, meaning that smaller modules have neurons closer together. Indeed, *Figure 2* shows that most of the connections have a weight close to or higher than the geometric mean, and they have short physical distances. This result indicates that high global efficiency is achieved by keeping modules dense and spatially compact.

Global efficiency is a measure of integration since neurons can reach each other by taking a few steps, this allows communication with less noise or attenuation, resulting in better signal transmission. At the same time, a shorter physical connection between neurons would also generate lower wiring costs within modules (*Betzel et al., 2017*; *Chen et al., 2006*; *Chen et al., 2013*). Overall, greater efficiency reflects better communication inside modules, but how do neurons self-organize to elicit that? Considering the silent synapses activation hypothesis, it is expected that the nearest neurons will make silent synapses first, once dendrites and axons would reach neighboring neurons first as they outgrow. Therefore, taking into account that neurite outgrowth is dependent on low $[Ca^{2+}]_i$ (*Kater and Mills, 1991*) it is expected that neuronal processes stop outgrowing and making connections as the synapses are activated. As a result of this, the probability of neurons making connections with nearby neurons would be expected to be higher than with distant neurons. This synaptic spatial specificity has already been reported in other brain regions such as the neocortex (*Holmgren et al., 2003*; *Ko et al., 2011*) and the auditory cortex (*Levy and Reyes, 2012*; *Oswald et al., 2009*). Even more interesting, (*Ko et al., 2011*) showed that not just the connection probability of nearby neurons is higher, but also that they have correlated functions. Regarding the number of neurons within modules, if time-correlated firing establishes an effective connection, we could expect that the density of edges would remain low. It could also explain why the edge density stops growing, and why not all nearby neurons are connected, once the firing rate of most of the neurons is low.

Indeed, although there is no overlap among modules in the effectively connected networks, *Figure 5a* shows that modules overlap spatially. This suggests that synchronous neurons are not necessarily adjacent in the cultures. Given the limitation of synchronization between low firing rate neurons, it is expected that the number of neurons within modules be also limited. We interpret this as

the importance of neuron firing rate heterogeneity, reported since the first recorded DIV even though neurons had few connections, to keep the size of modules low. Such a hypothesis fits the module definition since neurons within modules would be more synchronized with one another than with neurons in other modules, making more connections inside than outside the framework. The wiring cost results also suggested this effect. When compared to randomly connected networks, actual effective networks presented higher wiring costs inside modules, indicating a higher number of connections, and a lower value outside modules indicating the opposite.

In networks that are composed of local modules, it is expected to find nodes responsible for the integration of these frameworks. These hub nodes are more influential or essential to the network than others (*Sporns, 2015*). The heterogeneity of impact and overall functioning of nodes in a network is another topological property of complex networks. The results presented herein suggest that this framework is also presented in the effective networks since there is a maintained heterogeneity in the number of connections among neurons independent of the development period. This indicates that most of the neurons have fewer connections while there is a minority of high-degree neurons. The same occurs for the strength distributions as shown in *Figure 1f*. Our hubness definition considering node degree, connection strength, and node centrality showed a relationship between neuron rankings in these measures and the firing rate. This new concept indicates why these important neurons integrate modules. Considering the effective connection as a result of an LTP-like process, it is expected that neurons with a higher firing rate have a higher probability to synchronize with other neurons and have stronger connections with other high firing rate neurons. The multiplicative scaling of synapse weights is considered to have an important computation power by keeping at the same time the LTP and long-term depression (LTD) synapse effects as well as the firing rate of neurons within a stable range (*Turrigiano and Nelson, 2004*). The compensatory change of the input strengths to keep an overall resulting strength introduces competition between synapses.

If neurons with higher firing rates make stronger synapses, by competition, the synapses between neurons with a score of 4 would be privileged over synapses with lower firing rate neurons. Indeed, the $P_c$ result for neurons with a score of 4 is significantly higher than neurons with a score of 1, indicating their participation in other module connections. Additionally, we verified that the higher the score of a neuron the higher its median $P_c$, which suggests that the higher the score of a neuron the higher its firing rate and therefore the higher its likelihood to connect with high score neurons inside and outside its module. This hypothesis is corroborated by the rich-club results, which showed that neurons with a score of 4 comprised the largest fraction of higher weights in outside module connections when compared to neurons with a score of 1. The presence of neurons with a higher firing rate inside rich clubs was also reported by *Nigam et al., 2016*. They found that 'rich' neurons (considering all network connections in both in vivo and in vitro preparations) had on average two times the mean firing rate of the network.

*Figure 6b* shows that the neurons are equally divided among the scores in the interval of 21–27 DIV. Although only values for neurons with a score of 4 and a score of 1 were considered different for most of the results. Further, the significant difference between neurons with a score of 3 and neurons with a score of 1, as well as neurons with a score of 4 and neurons with a score of 2 in some DIV, suggests that the heterogeneity of the neuron role in a network is more complex than the definition of a hub and a non-hub node.

## Limitations

One of the most significant issues associated with inferring connectivity using in vitro preparations is the sub-sampling of the networks. The culture itself is comprised of thousands of neurons; however, we are only analyzing those neurons close to the 60 electrodes, which leave out the vast majority of active neurons in the culture. This likely leads to bias in computing topological measures. Despite this issue, the high degree of consistency across cultures suggests to us that the results are robust, if possibly biased in one direction.

Effective/functional connectivity inference may be influenced by heterogeneous coverage of neurons on the substrate (*Okujeni et al., 2017*; *Tibau et al., 2020*). *Timme et al., 2016b* coated themulti-electrode array (MEA) surface with polyethyleneimine (PEI) plus laminin when gathering the data set used in the present work. The PEI has shown to provide less clumping of neurons in the MEA

than using polylysine (*Hales et al., 2010*). However, using calcium imagine (with all neurons accessible) may improve the quality of the analysis and provide additional insights.

Another limitation is that individual neurons could not be tracked over multiple DIV, thus we had no control over which neuron was represented by each node for the same culture over time. Moreover, we were not able to identify which neuron was excitatory or inhibitory. Such distinction could elucidate the role of both types of neurons in the assembly self-organization. Mainly in the convergence, divergence, and recurrence of the information flow. Besides that, many of the concepts discussed herein were evidenced in excitatory synapses, however, little is known about their effect on inhibitory ones.

We formulated a hypothesis involving silent synapse activation and STDP mechanisms that might have a role in the activity-dependent self-organization of neural circuits. However, to provide insights into causal relationships between such hypothesis and integrations/segregation, modular organization, and important nodes, it would be interesting to construct a computational model showing the dependence of connectivity probability on Euclidean distance and that connections are established according to the synchronization of neurons. Another option is to apply electrical stimulation in neuronal cultures to induce control of the mentioned mechanisms.

## Future directions

While the emergence of complex networks may be partly explained by the role of spontaneous neural activity, it is still necessary to test the causal relations as our results are observational in nature. By manipulating the developing network with treatments such as drugs or optogenetic stimulation, we may be able to learn, not only how self-organization occurs naturally, but to control the formation of specific structures.

Besides the organization, the understanding of network navigation and how neuronal dynamics guide information flow by the topology are essential to fully comprehend how function and structure relate to each other, and how the network self-organizes and neurons communicate.

Finally, investigating whether these patterns also occur in circuits involving fundamental functions for example in central pattern generators may be important for generalizations of the process.

## Conclusion

Our findings suggest that plasticity and homeostatic mechanisms drive the emergence of segregated and integrated architectures in developing effective networks by reinforcing synchronized spontaneous activity. These processes induce a predictive relationship between the spike trains of pre- and post-synaptic neurons that produces reliable effective network patterns, such as the clustering of low firing rate neurons, the formation of modules, and the connection of high firing rate neurons across modules, integrating them. Such mechanisms, despite being independent of the exact physical location of each neuron, showed to have a preference to link neurons that are closer to each other. Finally, this organization involves a level of randomness, but it is greatly dependent on the heterogeneity of the firing rate of neurons.

# Materials and methods
## Data description

All animal care and treatment were done in accordance with the guidelines from the National Institutes of Health and all animal procedures were approved by the Indiana University Animal Care and Use Committee (Protocol: 11–041). The entire protocol description for data gathering and preprocessing is available in *Timme et al., 2016b* work. Briefly, female pregnant Sprague-Dawley rats were euthanized with $CO_2$ on gestational day 18 for embryo extraction. All embryos' hippocampi were combined for neuron dissociation. Dissociated neurons were plated in a density of 10.000 cells/μL (approximately $2.0 \times 10^5$ cells per culture) in an $8 \times 8$ square MEA (Multichannel Systems, 60 electrodes, 200 μm electrode spacing, and 30 μm electrode diameter). The cultures were recorded from 6 to 35 DIV, in intermittent time intervals for each culture. This implies that not necessarily the same cultures were compared over DIV. The data set totals 435 59 min recordings at a 20 kHz sampling rate. Most of the recordings have about one hundred neurons (min: 3, max: 142, mean: 91, total: 39,529). The average firing rate of the neurons was 1.9 Hz.

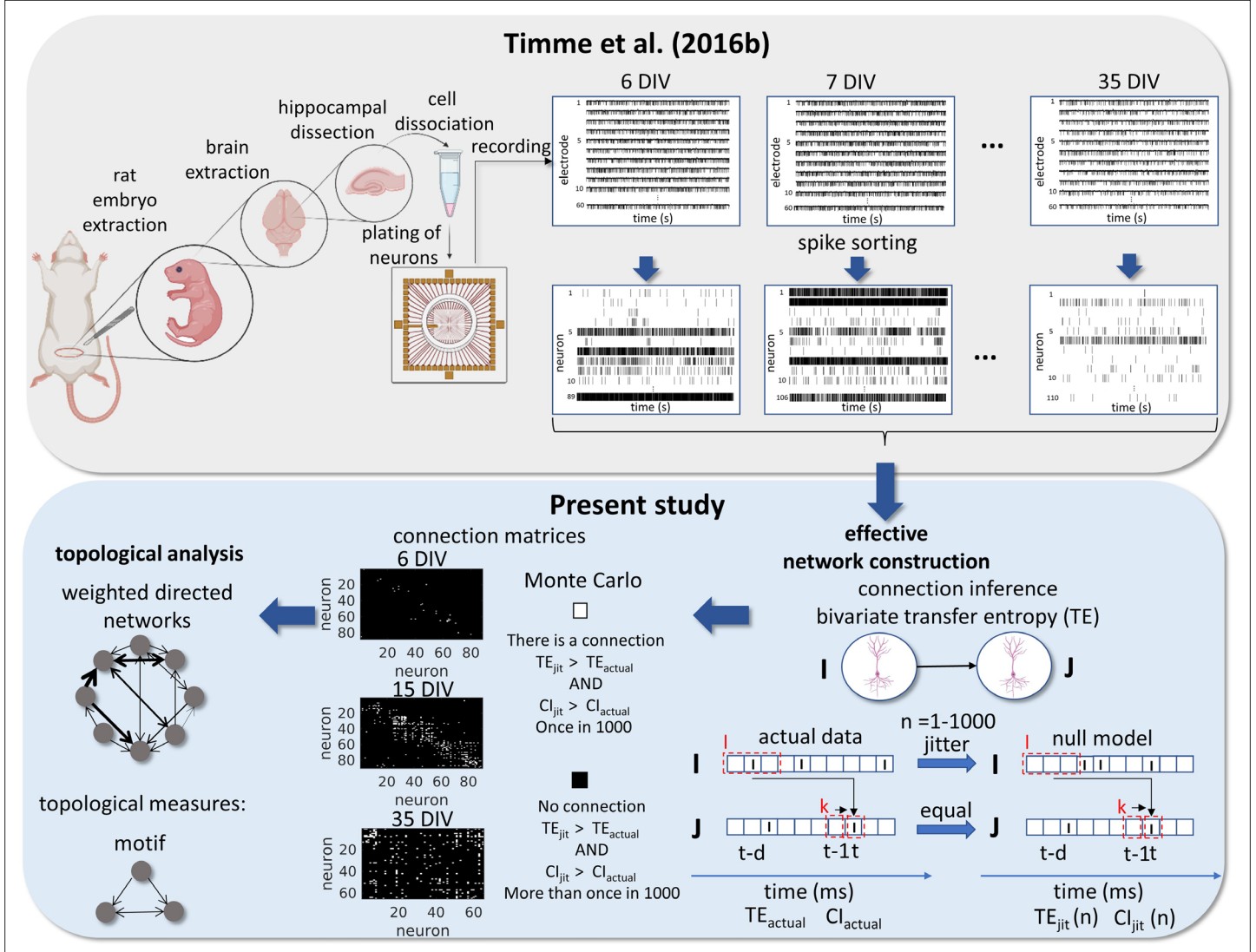

**Figure 8.** Experimental and data analysis procedure. Top row, left to right, bottom row, right to left. In the work of *Timme et al., 2016a* brains were extracted from embryos on gestational day 18. The hippocampi were dissected from the rest of the neuronal tissue and the hippocampal neurons were dissociated and plated on a multi-electrode array (MEA). The cultures were recorded from 6 to 35 days in vitro (DIV). The signals from the 60 electrodes were spike-sorted to reconstruct a binned and binary time series for each neuron. From this data set, we constructed effective networks, by applying a transfer entropy (TE) bivariate analysis. To infer a connection from a neuron I to a neuron J we computed the TE for each delay ranging from 1 to 20ms. Taking the highest value as the $TE_{actual}$ and the considered areas of the TE result curve in function of delays as the $CI_{actual}$. The time series of neuron I was jittered and the $TE_{jit}(n)$ and $CI_{jit}(n)$ were calculated following the same steps for actual data with n ranging from 1 to 1000. Connection matrices were constructed using a Monte Carlo approach by comparing how many times the TE results for the null model were higher than the result for actual data. The resulting weighted directed networks were constructed by normalizing the $TE_{actual}$ value of the connections considered significant by the entropy of the neuron J. Finally, we analyzed the progression of the topology of these networks over time.

Signal preprocessing was done offline. Individual neuron spikes were sorted by the wave_Clus spike sorting algorithm (*Quiroga et al., 2004*). Briefly, a threshold of 5 standard deviations was used to detect putative spikes. After detecting the spikes, waveforms were wavelet transformed and clusters were formed with the 10 most non-normally distributed coefficients. The spike sorted results were manually checked and the time stamps for each neuron were binned in 0.05ms bins. All experimental and data analysis procedures are represented in *Figure 8*.

## Effective networks inference

We constructed directed and weighted effective networks from binary spike trains for each putative neuron. Effective connectivity was inferred using TE. TE has been widely used as a network inference

methodology for discrete and continuous signals. Further TE has advantages when compared with other measures, such as being non-parametric, sensitivity to non-linear relationships, and can incorporate both excitatory and inhibitory connections. TE was first introduced by *Schreiber, 2000* as a measure of interaction between two time series: a source series I and a target series J. TE quantifies how much better an observer is at predicting the future of J after information from the future of I has been accounted for (above and beyond the information provided by I's statistics alone). Multiple time bins can be embedded in the analysis. *Ito et al., 2011* showed that to account for synaptic delays between neurons by using multiple bins of past history TE and considering message length higher than one bin is a robust methodology to predict the effective connectivity between neurons using discrete time series. The equation for TE considering multiple time delays and message length was given by:

$$TE_{I \rightarrow J}(d) = \sum_{j_t, j_{t-1}, i_{t-1}} p\left(j_t, j_{t-1}^k, i_{t-d}^l\right) log_2 \left( \frac{p\left(j_t | j_{t-1}^k, i_{t-d}^l\right)}{p\left(j_t | j_{t-1}^k\right)} \right)$$

where *d* was the multiple time delays from 0 to 20 ms, *k* is the number of bins of history from the receiver considered and *l* is the number of bins of history from the sender considered. The numbers used were 1 and 3 bins respectively since they provided the best performance found by *Ito et al., 2011*. After computing the TE for the range of delays for one pair of neurons we considered only the highest value.

Despite being a very used metric for effective connectivity inference TE is bivariate and has limitations that we had to compensate for to construct a robust analysis capable of excluding spurious connections. TE values are highly affected by the firing rate of neurons. As in this work, we are especially interested in information transfer by spike timing and not by spike rate; we controlled the firing rate influence by normalizing the TE values by the entropy of the receptor neuron (*Faber et al., 2018*; *Timme et al., 2016a*).

Further, dissociated neuron cultures present population bursts (*Maeda et al., 1995*; *Wagenaar et al., 2006*) that may elevate the firing rate of many neurons in a time interval. Such an effect can increase the TE value between these neurons. To address this issue, we used the TE as a function of the applied delays to detect whether the time dependence is because of bursts' influence or not. To control for the possibility of spurious connections due to the strict non-negativity of transfer entropy, we significance-tested each prospective edge using a Monte Carlo approach: jittering spikes to disrupt fine-grained temporal correlations, while approximately preserving the local firing rate and totally preserving the global firing rate. Finally, we tested our pipeline in a modeled network based on the Izhikevich algorithm. All these steps are better described in the following sections.

## Extraction of the influence of population bursts

Hippocampal dissociated neuron cultures as utilized in this work present intermittent and spontaneous synchronicity of electrical activity known as population bursts (*Penn et al., 2016*). Population bursts refer to a transient, global increase in the coherent activity of a large number of neurons, and can vary with the number of cells and development stage (*Wagenaar et al., 2006*). As we used cultures during the maturation process in this work, the electrophysiological activity had high variations and we felt the need for some type of stable and adaptive methodology to extract the influence of population bursts. *Beggs and Plenz, 2003* found that when plotting the TE values by multiple delays, the function tends to peak around the "natural" time delay of the two neurons producing a sharp distribution spanning around ~5 ms around this peak if the neurons are synaptically connected. If the connection is spuriously generated by population bursts, they observed a broad shape around 50–200 ms. We computed the TE value as a function of delays, and we divided the total area of the graph into two areas X and Y. Area X is defined as the 4 ms area around the TE peak and area Y as the rest of the area in the plot (*Ito et al., 2011*; *Nigam et al., 2016*). The coincidence index (CI) was used as a ratio between area X and Y to measure the sharpness of area X:

$$CI = \frac{X}{X+Y}$$

The closer CI is from 1 the sharper the peak.

Both high-order TE and CI were computed using the freely available TE toolbox developed by John Beggs' group (posted at: http://code.google.com/p/transfer-entropy-toolbox/; *Ito et al., 2011*).

## Significant connections test

Directed connections are expected to have both higher TE and CI values than by chance. In order to analyze the significance of the connections, we used a Monte Carlo approach. As a null model, we jittered the spike times only from the sender neuron to conserve the auto-prediction of the receiver neuron. This approach preserves the firing rate and forces the temporal correlation between spikes in the two time series to occur by chance. We jittered each spike in the source series by using a Gaussian distribution centered in the actual spike time and with a standard deviation of 10ms. The Gaussian distribution and the short standard deviation make the analysis stringent. We jittered I and computed TE and CI 1000 times for each pair of neurons. The connections were deemed significant if both TE and CI values calculated for the null model were higher than actual values up to once ($\alpha$=0.001).

## Transfer entropy normalization

In order to remove the firing rate influence on the connection weights we normalized the TE values by the entropy of the receiver neuron:

$$TE_{Norm,I \to J}(d) = \frac{\sum_{j_t j_{t-1}, i_{t-1}} p\left(j_t, j_{t-1}^k, i_{t-d}^l\right) log_2\left(\frac{p\left(j_t \vee j_{t-1}^k, i_{t-d}^l\right)}{p\left(j_t \vee j_{t-1}^k\right)}\right)}{-\sum_{j_t} p(j_t) log_2\left(p(j_t)\right)}$$

This normalization can be interpreted as the percentage of the receiver neuron entropy that can be accounted for by the sender neuron rather than the information transferred (*Faber et al., 2018*; *Timme et al., 2016a*).

## Validation of effective connectivity inference methodology

In order to verify if our pipeline is efficient in effective connectivity detection, we used a network model based on Izhikevich's model (*Izhikevich, 2006*).

The modeled networks had 800 excitatory and 200 inhibitory neurons. Each excitatory neuron was connected randomly to 100 neurons, resulting in a probability of connection equal to 0.1, while each inhibitory neuron was connected to 100 excitatory neurons only. Each excitatory-excitatory synapse had a delay between 0–20 ms while inhibitory-excitatory synapses had a delay of 1ms.

The dynamics of activation of each neuron was defined by a set of equations (*Izhikevich, 2003*):

$$v' = 0.04v^2 + 5v + 140 - u + I_{Syn}$$

$$u' = a\left(bv - u\right)$$

$$\text{If} \quad v\left(t\right) = 30mV, \text{ then} \quad v \leftarrow c \quad \text{e} \quad u \leftarrow u + d$$

where $v$ is the neuron voltage, $v'$ is the time derivative of the voltage, $u$ is a variable related to neuron recovery, $u'$ is the time derivative of the recovery variable, and $I_{Syn}$ is the total synaptic input received by the neuron, including a thalamic synapse delivered at random times following a Poisson process with an average rate of 1 Hz. The variables a, b, c, and d are adjustable parameters that govern the firing behavior depending on the type of neuron. The model was adjusted to have 800 regular spiking (RS) excitatory neurons, what means, (a, b, c, and d) = (0.02, 0.2, 265, and 8) and 200 fast-spiking (FS) inhibitory neurons, modeled by (a, b, c, and d) = (0.1, 0.2, 265, and 2). The connections weight was chosen as the starting parameters in the spiking-time dependent plasticity STDP Izhikevich's model (*Izhikevich, 2006*), 6 mV for excitatory neurons, –5 mV for inhibitory neurons, and 20 mV for thalamic synapses.

To approximate the model to the recordings from MEA we sub-sampled the model by randomly choosing only 80 excitatory and 20 inhibitory neurons from the entire network of 1000 neurons. The simulation was done for 30 min of recording, keeping the weight of the connections, once the STDP function was disabled in the model. The activation dynamics of the neurons were marked by the periodic occurrence of population bursts, portraying only the worst situation of the registered signals.

We applied the effective connectivity detection methodology previously described to the modeled data sets and compared the prediction of the effective connections with the synaptic connections

constructing a ROC curve based on the true and false positives (TP and FP, respectively) and negative predictions (TN and FN, respectively). Where the true positive rate (TPR) and the false positive rate (FPR) are given by the following equations:

$$TPR = \frac{TP}{TP+FN}$$

$$FPR = \frac{FP}{FP+TN}$$

## Network topological measures

After constructing the networks for each culture recording and validating the methodology we homogenize the networks. Firstly, by cutting off neurons with less than on average 5 spikes per second (0.2 Hz), considering that this number of spikes is not enough to calculate a reliable TE value. Then, by measuring the edge density of the networks from the normalization of the number of actual connections by the total number of possible connections ($N^2 - N$, where N is the total number of nodes; *Newman, 2003*). We selected only the networks with a density within the non-parametric CI with 95% confidence calculated using all the network densities. The density per period was defined as the median for each culture (28 cultures) over 3 days. We also cut off unconnected neurons.

The topology was analyzed from the resulting networks using standard or adapted graph-theoretic measures as described below. They were implemented by using the *Brain Connectivity Toolbox* (*Rubinov and Sporns, 2010*).

## Adapted measures
### Modules

In order to investigate the subdivision of the networks into modules we used an algorithm based on the multi-iterative generalization of the Louvain community detection algorithm (*Blondel et al., 2008*). Briefly, we first varied 1,000 times the resolution parameter (γ) in a log scale [$10^{-5}$, $10^5$] to find modules of many different sizes (*Betzel and Bassett, 2017*). Then, we selected a γ range where the number of modules ranged from 2 to the total number of nodes in the network. We utilized the extremities of this range to construct a new log scale of γ, including 1000 values. We then applied the new γ range to fine sample module detection within the just established limits. Finally, we used a consensus clustering algorithm (*Lancichinetti and Fortunato, 2012*) seeking a consensus partition of the module detection result. Where the agreement matrix resulting from the Louvain algorithm was thresholded at a level of 0.45 ($\tau$) to remove weak elements. The resulting matrix was partitioned again 10 times, using the Louvain algorithm with $\gamma$=1 (classic modularity). Finally, a new agreement matrix was built with the convergence of the partitions to one, representing the division of the network into modules.

### Hubness

To evaluate the presence of topologically important nodes in the networks we measured 4 parameters as indicators of node importance. Node **degree**, defined as the number of connections for each node. Node **strength**, the sum of the weights of the connections for each node. **Betweenness centrality**, the fraction of the shortest path length of the network that goes through the node (*Freeman, 1977*). Considering the idea that a node is central if it participates in most of the information that flows by the network. And **closeness centrality**, the distance (number of edges) that a node is from all the other nodes in the network. Calculated by using the inverse of the path length between a given node and all the other ones. A node with a higher closeness centrality value can achieve other nodes by shortest paths, which means it can exert a high influence on them (*Sporns, 2015*).

These measures were computed for each neuron recorded in the same culture on all DIV. Then, we pooled all the values together and classified neurons by score. If a neuron ranked in the 40% of the highest values for the 4 measures its score was 4. A score of 3 was attributed to neurons ranking in only 3 measures, and so on.

### Rich club

This coefficient represents the fraction of the highest weights of the network presented in a subnetwork (*Nigam et al., 2016*). Since we were interested in investigating the distribution of the highest connection weights between modules across nodes with different scores, we computed the rich club

coefficient considering only connections outside modules. The subnetworks are usually represented by nodes with different degrees; however, we adapted the (*Opsahl et al., 2008*) rich-club definition to consider scores as the richness parameter as follows:

$$\phi^W\left(score\right) = \frac{W_{score}}{\sum_{l=1}^{E_{score}} w_l^{ranked}}$$

where, $W_{score}$ is the sum of the weights of the connections outside the modules considering only the subset of nodes with a score higher than a given one, $E_{score}$ is the number of connections outside the modules considering the same subset of nodes, and $w_l^{ranked}$ is the ranked weights of all connections outside the modules (from largest to smallest).

### Spatial separation

As the interelectrode distance in the MEA is 200 μm, we were able to calculate the Euclidean distance between all the electrodes in the plate. Besides the timestamps, the data set used in this work has information about which electrode recorded the electrophysiological signal from each neuron. We used this information to calculate the mean distance of all connections within each module by computing the distance between the electrodes that gathered the signal from the two neurons involved in each connection.

### Wiring cost

We also used the Euclidean distance between the electrodes that gathered the signal from the two neurons involved in each connection to calculate the wiring cost. To compute the wiring cost inside modules we summed up the distance of all connections between neurons inside the modules. For the wiring cost outside modules, we summed up the distance of the connections between neurons in different modules. Both measures were normalized by the total wiring cost of all connections in the network.

## Standard measures

### Components

We used Dulmage-Meldensohn's decomposition to divide the networks into components by making a partition of the network nodes from a graph bipartition into connected subsets, then we analyzed how the number and size of components changed over DIV.

### Clustering coefficient

We computed the global clustering coefficient for each network by averaging the local clustering coefficient, defined as the number of a node's neighbors that were also connected, for all the nodes within the network.

### Motifs

We analyzed the 13 structural motif patterns presented by *Sporns and Kötter, 2004*, which form a basic structural alphabet of connections considering 3 nodes. The frequency in which each of the 13 patterns emerged in every network during maturation was measured.

### Path length

Since we used weighted networks, we computed the path by averaging the sum of edge lengths in the network, defined as the inverse of the edge weight.

### Global efficiency

We computed the global efficiency within a module as the average efficiency between all node pairs in that module.

### Small-worldness

The small-world architecture is a common organization of complex networks that rovers between regular and random organizations. As it encompasses highly clustered nodes with small characteristic

path lengths, we measured the small-worldness in comparison with random networks, that are known to be poorly clustered and have a small average path length (*Watts and Strogatz, 1998*):

$$S = \frac{\frac{C}{C_{random}}}{\frac{L}{L_{random}}}$$

where $C$ and $L$ are the global clustering coefficient and path length, respectively, for actual data e $C_{random}$ and $L_{random}$ are the same coefficients calculated for random networks (see Null models below for additional details).

## Participation coefficient

Providing information about the contribution of individual nodes in connections to other modules than its own module we computed the participation coefficient for nodes with different scores as follows:

$$Pc_i = 1 - \sum_{s=1}^{N_m} \left( \frac{K_{is}}{k_i} \right)^2$$

where $N_m$ is the number of modules, $K_{is}$ is the number of links that neuron $i$ makes with module $s$, and $k_i$ is the total degree of node $i$. If the links of node $i$ are uniformly distributed by all modules $Pc_i = 1$. Conversely, if all the links of node $i$ are within its own module $Pc_i = 0$ (*Guimerà and Nunes Amaral, 2005*).

## Null models

These network topological measures can vary significantly according to the edge density. As we are analyzing the self-organization process during maturation, it is expected that the networks have different densities. Besides that, we would like to study whether self-organization has a random or a complex influence. Therefore, we compare the results of the measures previously described for the actual networks with a null model by averaging the outcome of the measures from 100 random networks, constructed by randomly rewiring each edge approximately 20 times. The actual number of in and out connections (in- and out-degree respectively) distributions and the number of reciprocal connections were preserved in the null model.

## Statistical analysis

The firing rate distributions of all recorded neurons and the weight distribution of all persistent connections for each DIV were fitted by the generalized Pareto probability density function (Matlab R2022a) while the values in a log scale were fitted by the normal probability density function. The log-normal behavior of the distributions was tested by using a Lilliefors test, alpha of 0.01.

The number of neurons by module distribution was fitted by the Poisson probability density function as follows:

$$f\left( x \vee \lambda \right) = \frac{\lambda^x}{x} e^{-\lambda}; x = 1, 2, 3, ..., \infty.$$

where $\lambda$ is the variance of the Poisson distribution.

The degree and strength distributions were built by counting the probability of the degrees and strength, respectively, within a range for each DIV.

The weight vs distance probability analysis was performed by a bivariate histogram normalized by the sum of the total number of observations. The mean probability density of connections with different weights over distance and the probability of connections with different weights for distances of 25 and 1465 μm were tested by using the Kruskal-Wallis test followed by the Tukey-Kramer post-hoc test and alpha of 0.05, for each DIV.

Among DIV or score comparisons for density, number of modules, rich-club coefficient, firing rate, and $P_c$ were made by using the Kruskal-Wallis test followed by the Tukey-Kramer post-hoc test and an α of 0.05.

To test whether the number of components and small-worldness, as well as the normalizations of the clustering coefficient, path length, and motifs, come from a population with a median equal to 1 the Wilcoxon signed-rank test was used, with an α of 0.05.

The correlation between the module global efficiency and the total number of neurons per module, as well as the mean spatial separation and the total number of neurons per module, was computed by using Spearman's rho.

Since the wiring cost and the fraction of nodes by the score are fractions, their results were compared on the same DIV by using the ANOVA and the one-way ANOVA tests, respectively, with an $\alpha$ of 0.05.

## Acknowledgements

We would like to gratefully acknowledge the alumni and current members of John Beggs's lab at IU Bloomington for the signal recordings and the toolboxes freely available. We also thank Daniel Pinheiro for the insightful discussions during the development of this work. This study was financed in part by the Coordenação de Aperfeiçoamento e Pessoal de Nível Superior - Brazil (CAPES) - Financial Code 001, and in part by the São Paulo Research Foundation (FAPESP), process 2018/12605–8. This research was supported in part by Lilly Endowment, Inc through its support for the Indiana University Pervasive Technology Institute.

## Additional information

### Funding

| Funder | Grant reference number | Author |
|---|---|---|
| Coordenação de Aperfeiçoamento de Pessoal de Nível Superior | Financial Code 001 | Priscila C Antonello |
| Fundação de Amparo à Pesquisa do Estado de São Paulo | process 2018/12605-8 | Marimélia Porcionatto Jean Faber Priscila C Antonello |

The funders had no role in study design, data collection and interpretation, or the decision to submit the work for publication.

### Author contributions

Priscila C Antonello, Conceptualization, Data curation, Formal analysis, Funding acquisition, Investigation, Methodology, Software, Visualization, Writing – original draft, Writing – review and editing; Thomas F Varley, Methodology, Writing – review and editing; John Beggs, Data curation, Methodology, Writing – review and editing; Marimélia Porcionatto, Conceptualization, Funding acquisition, Project administration, Supervision, Writing – original draft; Olaf Sporns, Conceptualization, Data curation, Formal analysis, Investigation, Methodology, Project administration, Resources, Software, Supervision, Validation, Visualization, Writing – original draft; Jean Faber, Conceptualization, Formal analysis, Funding acquisition, Investigation, Methodology, Project administration, Resources, Supervision, Validation, Visualization, Writing – original draft

### Author ORCIDs

Priscila C Antonello (iD) http://orcid.org/0000-0002-0624-1169
Thomas F Varley (iD) http://orcid.org/0000-0002-3317-9882
Jean Faber (iD) http://orcid.org/0000-0002-2129-8251

### Ethics

All animal care and treatment were done in accordance with the guidelines from the National Institute of Health and all animal procedures were approved by the Indiana University Animal Care and Use Committee (Protocol: 11-041).

### Decision letter and Author response

Decision letter https://doi.org/10.7554/eLife.74921.sa1
Author response https://doi.org/10.7554/eLife.74921.sa2

## Additional files

### Supplementary files
- Transparent reporting form

### Data availability

All data analyzed in this study is freely available on the Collaborative Research in Computational Neuroscience (CRCNS) data sharing initiative at http://doi.org/10.6080/K0PC308P.

The following previously published dataset was used:

| Author(s) | Year | Dataset title | Dataset URL | Database and Identifier |
|---|---|---|---|---|
| Timme NM, Marshall N, Bennett N, Ripp M, Lautzenhiser E, Beggs JM | 2016 | Spontaneous spiking activity of thousands of neurons in rat hippocampal dissociated cultures | http://doi.org/10.6080/K0PC308P | Collaborative Research in Computational Neuroscience, 10.6080/K0PC308P |

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
