## [Editor Report]

This paper investigates the emergence of complex network organization in neuronal circuits grown in vitro. Network analysis of neuronal activity recordings allowed a detailed assessment of how neurons self-organise into clusters of functionally segregated models while also retaining a capacity for integrated communication through a subset of highly active neurons. This work is of interest to researchers working on neuronal connectivity, brain development, and self-organisation in complex systems.

---

## [Decision Letter]

**Decision letter after peer review:**

Thank you for submitting your article "Self-organization of in vitro neuronal assemblies drives to complex network topology" for consideration by *eLife*. Your article has been reviewed by 3 peer reviewers, and the evaluation has been overseen by a Reviewing Editor and Michael Frank as the Senior Editor. The following individual involved in review of your submission has agreed to reveal their identity: Jordi Soriano (Reviewer #1).

Essential revisions:

1) Please clarify the following experimental details:

– According to Results and Methods, there were 424 networks used, which comprised from DIV 6 to 35. However, it is not clear whether the same cultures were present in the different days of analysis. For instance, some DIVs could include networks that were not present in the others. Since there at least 9 networks in each DIV, do these 9 correspond to 9 cultures that are present in all the other days?

– Please clarify the influence of the degree of neuronal aggregation in the neuronal cultures, i.e., whether neurons cover homogeneously the substrate or they group in islands. Is such information available? Aggregation could locally increase the density of neurons and potential connections. Different experimental studies have indeed shown that fluctuations in spatial neuronal density substantially affect network development and effective/functional connectivity traits (Okujeni et al., J Neurosci 2017; Tibau et al., IEEE Trans Net Sci Eng 2020). If the information is not available, the authors could mention such a limitation, and even explain that calcium imaging data (with all neurons accessible) could provide additional insight.

– From the spacing between electrodes, one gets about 1.5x1.5 mm^2 total square area. The lateral size of the square is of the order of the axonal length of neurons grown in culture, so potentially any neuron can easily connect with any other in the culture, making the approximation of a random connectivity in the simulations valid. This should be discussed. There are indeed works pointing out the importance of spatially embedding in cultures (e.g., Hernández-Navarro et al., Phys Rev Lett 2017), in which metric correlations can be neglected in small networks.

2) The authors seem to use the "effective connectivity validation" analysis to associate the inferred effective connections with structural ones, or at least a substantial part of them. This is a strong assumption that is later treated in the "Neuronal networks self-organization" section of the Discussion. However, it can be confusing when reading the Results, so an earlier clarification (in Results) of the limitations of effective connectivity inference is necessary, and the authors should explain that 'connections' in the article reflect strong paths for information flow rather than actual structural connections. Indeed, not all possible dynamical states of the network are present in a raster plot that portrays solely spontaneous activity; i.e., that information flow (from which effective connectivity is extracted) does not explore all possible structural paths. With inhibition active, several communication paths may be inactive or silent, although structural connections may exist. The extreme difficulty of inferring structure from dynamics has motivated experimentalists to compare effective connectivity inferred from evoked activity with spontaneous activity, to later compare with physiological information (see e.g., Bauer et a., Cerebral Cortex 2018), observing that evoked activity better captures the network's underlying circuitry.

3) Related to the above, it would be useful for the authors to run additional simulations of the same network with and without inhibition (i.e., by completely silencing the inhibitory neurons) and compare how many connections in the excitation-pure network are present in the excitation-inhibition one. A test relative to non-random graphs would also be helpful; in particular, spatial graphs in which connectivity probability depends on Euclidean distance (Orlandi et al., Nat Phys 2013) would be useful, and even help construct a model to explain the overall results, particularly in understanding that nearby neurons shape spatially compact communities.

4) Figures 1e shows the distribution of connections. The distribution shifts to higher k values as maturation progresses, indicating an increase of connectivity along development. The authors should include an inset showing the p(k) value for a given k (e.g., k=10, 20, and 40) to illustrate that p(k) gradually goes up.

5) Figure 1g shows the evolution of Euclidean connectivity distances along DIV. This is an important result since it illustrates the gradual evolution from a segregated to an integrated network. The authors could place the 'probability' color bar at the top of the figure in a horizontal manner and leave the bottom-right corner to plot the average connectivity distance as a function of DIV. Additionally, the panel of Figure 1g contains the results averaged over recordings. Can the authors show as a supplementary figure the evolution of the same network along DIV?

6) Neuronal circuits in vivo and in vitro experience GABA switch (Soriano et al., PNAS 2008; Tibau et al., Front Neur Circ 2013; Tibau et al., IEEE Trans Net Sci Eng 2020) in which inhibition behaves as excitatory up to DIV 7-8, and afterwards has its normal inhibitory role. In panel 1g, connections seem to extend a larger distance at DIV 6 than at DIV 9, and I think GABA switch is the explanation. At DIV 6, GABA structural connections (behaving as excitatory) could extend longer distances than the excitatory ones and lead to the emergence of long-distance effective connections, which suddenly vanish at DIV 9. If so, this could be explained in the discussion, also addressing the fact that GABA switch is not analyzed in this work. The authors address GABA switch in line 386, but they could extend the discussion a bit more.

7) Please clarify why most of the motifs (Figure 2f) to drop after DIV 27?

8) Many neural mechanisms including silent synapses and STDP are discussed in this manuscript. However, neither generative network model nor neural circuit model is developed to directly illustrate possible mechanisms underlying network formation and development. In the absence of a generative model providing insights into causal relationships between the modular organization, neuronal hubs and segregation/integration, the authors should clear state that their discussion of mechanisms is primarily speculative, and suggest strategies for gaining further mechanistic insight.

9) Effective connectivity is inferred by using a transfer entropy analysis of electrophysiological signals. However, it has been questioned that the transfer entropy (James, et al., 2016) does not quantify the flow of information as commonly assumed. Given this concern, some clarifications or comparisons with state-of-art methods need to be provided.

10) What is the relation between the log-normal distribution of coupling weights and that of firing rate? What do these heavy-tailed distributions mean for collective network states and whether they are related to the key connectivity properties such as neuronal hubs? Given that all these phenomena have been reported in previous studies, without an in-depth investigation of the problems, the novel contribution of this study is somewhat unclear to me.

11) In Figure 1c and Figure 1d, the log normal distribution is used to fit the data. Is this distribution better than other heavy-tailed distribution such as truncated Pareto distribution?

12) Line 88 – "the term neuronal network to refer to the neuronal assemblies coupled to electrodes and effective networks to refer to the network inferred from the neural recordings". This sentence is not correctly highlighting these two central concepts of the paper. Considering what was written a few lines above, and by also looking at the Results and Methods sections, maybe it would be easier for the reader to follow something along these lines: "the term neuronal networks to refer to transiently coupled neuronal assemblies and effective networks to refer to the circuits inferred from neural recordings". Of course, this is just a suggestion. The aim is to make as clear as possible the intentions of the authors to disentangle a dynamical aspect ('neuronal assembly') and a structural aspect ('effective network').

13) Line 145 – "Only some connections extended for long distances, and these connections were more likely to also have a weight close to the geometric mean. Connections with lower and higher weight values were more likely to have shorter lengths." Very interesting! Can the authors provide a statistical characterization of these effects and a corresponding figure? In particular it is important to test if this feature is also present in the two other models (Watts-Strogatz and Kwok et al.). If yes, it should be mentioned. If not, then it is an important distinguishing feature, and in this case, this aspect should be considerably expanded.

14) Line 290 – "the higher the neuron score, the higher the median firing rate over all DIVs." The authors took care of choosing the best inferring method available (transfer entropy). However, a big issue of the structure-from-dynamic approach is its circularity. Dynamical information is used to infer the structure of a network and to draw conclusions concerning how the dynamically-derived structure affects network dynamic itself. Therefore the correlation between firing rate and neuron 'hub' scoring (sentence at the beginning of this paragraph) more than being a result of the study might be a side-effect of the analysis approach. The authors already took the care of searching the available literature to find support (like Sung et al. 2005 for the firing rates of neurons), they are invited to expand their discussion on this point, and suggest experimental and analytical solutions, e.g. use electron microscopy to validate (at least a portion of) the activity-derived networks, or the literature they found to also compare their motifs results, or, in case, why their results cannot be compared with the available literature and electron microscopy data. The use of a spiking network to test the validity of the inference procedure is interesting and valid as a preliminary test, but would require an entire separate work to be able to use it to break the structure-from-dynamic circularity. So this point too needs to be clearly discussed.

15) Line 350 – Even though it is the discussion, it should be clearly indicated that by "self-organization", in the context of the experimental strategy and presented results, the authors cannot mean functional connectivity (as it is hinted by the first paragraph, line 352-257), but only self-organization as self-reinforcement of synaptic connections between neurons with correlated firing (as it is done in the following paragraphs of discussion). If, for example, spatially organized stimuli through the electrode were used, and effects on the network structure were observed, it could have been possible to advance the idea that the network self-organization would be functional. In any case, the silent synapses hypothesis they suggest is fundamental to understand *also* functional results. They could maybe move the first paragraph to the end?

16) Line 62 "The spontaneous emergence of spatio-temporal patterns driven by neuronal activity is characteristic of a self-organizing system." maybe could be a little expanded given that the rest of the article will be a balance of structure and dynamic running over it.

17) Line 75 "DIV" is first used without being defined. The definition (days in vitro) comes for the first time in the caption of figure 3 (line 1211)

18) Line 194, "only 5 of the 13 patterns (5, 8, 11, 12, and 13), were", the comma after the right bracket should be dropped.

19) Line 216 "Subgraphs", "modules", and "communities" seem to be used interchangeably. However they underlie different meanings (respectively theoretical, architectural, and functional). Wouldn't it be easier for the reader to stick to one term, if just one meaning is intended, or explain the term and its use, where appropriate?

The clustering grows towards the 12 DIV then decreases, as path length and small-worldness (Figure 2cde). Is this backed up by other data? We are in use with the idea that brain networks are small-world, it looks like these networks, towards the 20 DIVs are not.

20) Line 290, "Figure 4d shows that the higher…". It should be "Figure 4e". Additionally, for clarity in the presentation of results, panel 4f (participation coefficient) should go above panel 4e (firing rate).

21) Line 346 "neuron loss drastically impacted the topological organization of the effective networks, resulting in a breakup of the networks into different components" In addition (and before) pathological conditions, the neuron loss can also be a (functional) feature of the multi-stage process of circuit refinement, as the authors found in the formation of communities.

22) Lines 428,429 There is a list of properties but is written in a series of sentences (lacking verbs). "… Strogatz, 2001). A short path …" should be ".. Strogatz, 2001), a short path …". One line below, "… Ottino, 2004). And the presence …" should be: "… Ottino, 2004), and the presence …"

23) Line 774, the fonts for variables and text are the same, making difficult to read the equation.

---

## [Author Response]

Essential revisions:1) Please clarify the following experimental details:– According to Results and Methods, there were 424 networks used, which comprised from DIV 6 to 35. However, it is not clear whether the same cultures were present in the different days of analysis. For instance, some DIVs could include networks that were not present in the others. Since there at least 9 networks in each DIV, do these 9 correspond to 9 cultures that are present in all the other days?

The same culture was recorded for many DIV, although not for all of them. So yes, the same culture was present on different days of analysis. However, for each DIV different cultures could be considered. Therefore, the statistical analysis was made in an unpaired way.

We adjusted the following phrase in lines 713-715:

“The cultures were recorded from 6 to 35 DIV, in intermittent time intervals for each culture. This implies that not necessarily the same cultures were compared over DIV. The data set totals 435 59-minute recordings at a 20 kHz sampling rate.”

– Please clarify the influence of the degree of neuronal aggregation in the neuronal cultures, i.e., whether neurons cover homogeneously the substrate or they group in islands. Is such information available? Aggregation could locally increase the density of neurons and potential connections. Different experimental studies have indeed shown that fluctuations in spatial neuronal density substantially affect network development and effective/functional connectivity traits (Okujeni et al., J Neurosci 2017; Tibau et al., IEEE Trans Net Sci Eng 2020). If the information is not available, the authors could mention such a limitation, and even explain that calcium imaging data (with all neurons accessible) could provide additional insight.

Thank you for this pertinent comment.

Indeed, dissociated neurons tend to clump when in culture. Unfortunately, evidence of homogeneous neuron coverage in the cultures is not available in the data set. However, since the authors used the same protocol as Hales et al. (2010, doi: 10.3791/2056) and coated the MEA surface with polyethyleneimine (PEI) plus laminin, it indicates that neurons covered homogeneously the substrate. Hales et al. (2010; doi:10.3791/2056) reported that PEI provided less clustering of cells in the MEA than using polylysine. Besides that, in Figure 1A showed in Timme et al. (2016, 10.3389/fphys.2016.00425), work that reported the gathering of the data set used in this paper, neurons seem to be homogeneously distributed by the MEA. In contrast, we agree that, for example, calcium imagine would improve the quality of the analysis. Therefore, we have inserted the following paragraph in the "Limitations" section of the Discussion, lines 648-654:

“Effective/functional connectivity inference may be influenced by heterogeneous coverage of neurons on the substrate (Okujeni et al., 2017; Tibau et al., 2020). Nicholas M. Timme et al. (2016b)coted the MEA surface with polyethyleneimine (PEI) plus laminin when gathering the data set used in the present work. PEI has shown to provide less clumping of neurons in the MEA than using polylysine (Hales et al., 2010). However, using calcium imagine (with all neurons accessible) may improve the quality of the analysis and provide additional insights.”

– From the spacing between electrodes, one gets about 1.5x1.5 mm^2 total square area. The lateral size of the square is of the order of the axonal length of neurons grown in culture, so potentially any neuron can easily connect with any other in the culture, making the approximation of a random connectivity in the simulations valid. This should be discussed. There are indeed works pointing out the importance of spatially embedding in cultures (e.g., Hernández-Navarro et al., Phys Rev Lett 2017), in which metric correlations can be neglected in small networks.

Thank you for your suggestion.

Indeed, we totally agree with this possible issue. We insert the following paragraph in the discussion, lines 466-472:

“The total square recording area formed by the electrodes corresponds to about 1.5 x 1.5 mm². The lateral size of the square corresponds to the axonal growth of neurons in culture(Kaneko and Sankai, 2014), which, in principle, could foster a neuron to connect with any other in the culture. However, when we compared the topological measures for inferred networks with random networks the results were significantly different, showing the emergence of non-random topological properties and neurons’ tendency to connect with their neighbors.”

2) The authors seem to use the "effective connectivity validation" analysis to associate the inferred effective connections with structural ones, or at least a substantial part of them. This is a strong assumption that is later treated in the "Neuronal networks self-organization" section of the Discussion. However, it can be confusing when reading the Results, so an earlier clarification (in Results) of the limitations of effective connectivity inference is necessary, and the authors should explain that 'connections' in the article reflect strong paths for information flow rather than actual structural connections. Indeed, not all possible dynamical states of the network are present in a raster plot that portrays solely spontaneous activity; i.e., that information flow (from which effective connectivity is extracted) does not explore all possible structural paths. With inhibition active, several communication paths may be inactive or silent, although structural connections may exist. The extreme difficulty of inferring structure from dynamics has motivated experimentalists to compare effective connectivity inferred from evoked activity with spontaneous activity, to later compare with physiological information (see e.g., Bauer et a., Cerebral Cortex 2018), observing that evoked activity better captures the network's underlying circuitry.

Thank you for your comments and suggestions.

Indeed, this is a very important issue, and it deserves more clarification.

We believe part of this misunderstanding is due to the comparison of our experimental results with the computational results. Indeed, in Izhikevich’s model, the network dynamics is constructed based on the synaptic weights and delays between neuron activity propagation. In this way, the model reflects the information flow pathways based on structural connections. However, our results from experimental data could reflect only information flow from effective connections. Furthermore, we did not intend to prove the accuracy of the methodology in inferring structural connections through effective ones.

We used the computational model to test the ability of the pipeline in detecting the predictive information transfer. Nonetheless, we agree that our “connection” definition was confusing. We also agree that the limitations in inferring effective connections, as well as in TE analysis need to be better discussed. We have therefore added the following paragraph to the section “Validation of effective connectivity” in the Results section, lines 163-189:

“In our analysis, since we have access only to neuron spiking activity, the term “connections” reflects strong paths for information flow rather than actual structural connections (Lizier and Prokopenko, 2010). This type of inference in neural systems should be done carefully because of the number of influences that the analysis can have. Additionally, the relationship between the structural and effective connections may not fully match. Since the predicted transfer is computed from spike trains, not all possible dynamical states are presented in the spontaneous activity. With inhibition active, several communication paths may be inactive or silent, although structural connections may exist (Park and Friston, 2013). James, Barnett, and Crutchfield (2016) pointed out some limitations of TE in detecting information flow, they showed through simple examples how TE can overestimate flow and underestimate the influence. On the other side Garofalo et al. (2009), Ito et al. (2011) and Orlandi et al. (2014) showed how using a combination of time delays and adequate binning may overcome these limitations and make TE analysis a powerful tool for inferring effective connections. Furthermore, Nigam et al. (2016b) discussed the importance of handling firing rate and population bursts influence, as well as spurious connections caused by common drive and transitivity to have a more accurate effective connections inference.

Underpinned by these works we constructed the present methodology. To test the accuracy of the effective connectivity inference we used Izhikevich’s network model (Izhikevich, 2006). The network dynamics was constructed based on the synaptic weights and delays between neuron activity propagation. The model reflects information flow pathways based on structural connections between cortical excitatory and inhibitory neurons. The mean firing rate of excitatory neurons was 4.78 ± 0.84 Hz (mean ± SD) and of inhibitory neurons was 16.82 ± 2.19 Hz (mean ± SD). This model helps us to test the ability of the pipeline in detecting the predictive information transfer. Since we compared the transferred information computation with the structural couplings of neurons that generated the information transfer.”

3) Related to the above, it would be useful for the authors to run additional simulations of the same network with and without inhibition (i.e., by completely silencing the inhibitory neurons) and compare how many connections in the excitation-pure network are present in the excitation-inhibition one. A test relative to non-random graphs would also be helpful; in particular, spatial graphs in which connectivity probability depends on Euclidean distance (Orlandi et al., Nat Phys 2013) would be useful, and even help construct a model to explain the overall results, particularly in understanding that nearby neurons shape spatially compact communities.

Indeed, many works have been using evoked activity to better capture the network's underlying circuitry. However, since our analysis considered only baseline activities is expected that the activities and paths revealed by evoked stimuli could be radically different and hard to compare. Additionally, evoked activity may activate communication paths that were not active during the 1 hour of spontaneous activity recording which could yield different topological patterns. Furthermore, since we are using a bivariate theoretic information metric to compute the effective connections, if a neuron A fires and neuron B does not, as in an inhibitory synapse, this effect will decrease the uncertainty of the neuron B activity prediction considering the past activity of neuron A. In this way, the metric is also capable to capture inhibitory connections if the dynamics between A and B exist.

Nonetheless, we agree that the use of a computational model is a good way to get evidence. However, the suggested alterations in the model have some execution issues. When we silenced inhibitory neurons in the same networks used previously, the firing rate of excitatory neurons increase a lot going from on average ~5 Hz to ~350 Hz. This huge rise in firing rate produces an important technical issue since it took at least 5x more time to run the same model. We calculated that to generate 30 minutes of spike trains and compute the transfer entropy of these models, a process that also took a lot of time to run, it would spend about 1 month for each data set. Considering that we used 8 sets in our results, this new code would take around more than 8 months, at least, to run. As an alternative, we tried to use only 5 minutes of spike trains modeled, but the inferred network when compared with the modeled networks resulted in an area under the ROC curve equal to 0.5, showing the lack of accuracy to infer the networks. This result is probably due to the high firing rate, which affects the effectiveness of the null model in assisting the non-significative connections cut off. To make possible the use of only 5 minutes of the modeled high firing rate activity we would need to restrict the conditions to construct the null models, which will probably modify the results. Further, we also considered making a model where connectivity probability depends on Euclidean distance, which we agree would be very important to test our silent synapses hypothesis. Most of the time that we took to answer this revision was used by thinking, discussing, and running tests related to these models. However, we concluded that changing the model conditions would take a lot of time to build, run and run the transfer entropy analysis, which would use a lot more resources than was allocated to the execution of this work. In this way, we believe that the extension of the model to provide evidence other than to answer the central questions of this work would be very interesting and useful in further work.

We added the paragraph below in lines 663-670, addressing these considerations.

“We formulated a hypothesis involving silent synapse activation and STDP mechanisms that might have a role in the activity-dependent self-organization of neural circuits. However, to provide insights into causal relationships between such hypothesis and integrations/segregation, modular organization, and important nodes, it would be interesting to construct a computational model showing the dependence of connectivity probability on Euclidean distance and that connections are established according to the synchronization of neurons. Another option is to apply electrical stimulation in neuronal cultures to induce control of the mentioned mechanisms.”

4) Figures 1e shows the distribution of connections. The distribution shifts to higher k values as maturation progresses, indicating an increase of connectivity along development. The authors should include an inset showing the p(k) value for a given k (e.g., k=10, 20, and 40) to illustrate that p(k) gradually goes up.

Thank you for this comment and request.

We added an inset in Figure 1e showing values of P(k) for k=10,20,40 for all the DIV used in the analysis. The paragraph below was also added in the Results section, lines 146-148:

“The distributions shifted to higher values of degree as maturation progressed, indicating an increase in connectivity along with development (Inset of Figure 1e).”

5) Figure 1g shows the evolution of Euclidean connectivity distances along DIV. This is an important result since it illustrates the gradual evolution from a segregated to an integrated network. The authors could place the 'probability' color bar at the top of the figure in a horizontal manner and leave the bottom-right corner to plot the average connectivity distance as a function of DIV. Additionally, the panel of Figure 1g contains the results averaged over recordings. Can the authors show as a supplementary figure the evolution of the same network along DIV?

Thank you.

We expanded Figure 1g to Figure 2 and Figure 2 —figure supplement 1. We believe the suggestions are now covered in these two figures.

6) Neuronal circuits in vivo and in vitro experience GABA switch (Soriano et al., PNAS 2008; Tibau et al., Front Neur Circ 2013; Tibau et al., IEEE Trans Net Sci Eng 2020) in which inhibition behaves as excitatory up to DIV 7-8, and afterwards has its normal inhibitory role. In panel 1g, connections seem to extend a larger distance at DIV 6 than at DIV 9, and I think GABA switch is the explanation. At DIV 6, GABA structural connections (behaving as excitatory) could extend longer distances than the excitatory ones and lead to the emergence of long-distance effective connections, which suddenly vanish at DIV 9. If so, this could be explained in the discussion, also addressing the fact that GABA switch is not analyzed in this work. The authors address GABA switch in line 386, but they could extend the discussion a bit more.

Thank you for this important comment.

Indeed excitatory-inhibitory GABA switch is present during the neuronal assembly development period observed. However, as we are looking at information flow circuit formation is difficult to extend the discussion about this phenomenon without further evidence of its role in the process. Additionally, since we are using a bivariate theoretic information metric to compute the effective connections, if a neuron A fires and neuron B does not, as in an inhibitory synapse, this effect will decrease the uncertainty of the neuron B activity prediction considering the past activity of neuron A. In this way, the metric is also capable to capture inhibitory connections if the dynamics between A and B exist. Despite the difficulty in distinguishing which connection is excitatory and which connection is inhibitory (Garofalo et al. 2009, 10.1371/journal.pone.0006482, and Orlandi et al. 2014, 10.1371/journal.pone.0098842) Ito et al. (2011, doi:10.1371/journal.pone.0027431) showed an acceptable performance of transfer entropy in detecting inhibitory connections. This suggests that despite GABA influence being excitatory or inhibitory the communication path established by this neurotransmitter can be detected. The new Figure 2b e and Figure 2 —figure supplement 1c show that the shorter distances are more likely both in 6 and 9 DIV. The longer connections that are visualized in the 6 DIV distributions that are not present in the 9 DIV distributions may be a result of some specific culture that was recorded in the 6 DIV but was not recorded in 9 DIV, which would configure a sampling variation. This variance is also present in Figure 2e in the distance over the DIV plot (mean ± SEM). Figure 2 —figure supplement 1 shows the evolution of the joint probability distribution for the same network (Culture 1), we can notice that longer connections are emerging over time.

7) Please clarify why most of the motifs (Figure 2f) to drop after DIV 27?

Thank you for your question.

Since we did not make the experiments is hard to know exactly what might have happened. However, from the experiments done in our lab (not reported in this work), most probably this effect might be explained by neuronal death associated with the difficulties of maintaining a healthy cell culture beyond 4 weeks (Kaech and Banker, 2006, doi:10.1038/nprot.2006.356). Indeed, Figure 1b shows a decrease in the number of recorded neurons, although no significant decrease in the number of network edges has been detected (Figure 2a). The impact on the topology after such a period also includes a breakup of the networks into different components, as can be seen in Figure 2b. We also believe that the silencing of neurons might be a result of the multi-stage process of circuit refinement during development. But to test this hypothesis further in vivo studies should be performed. This discussion is presented in the “Changes over time” section in the Discussion.

8) Many neural mechanisms including silent synapses and STDP are discussed in this manuscript. However, neither generative network model nor neural circuit model is developed to directly illustrate possible mechanisms underlying network formation and development. In the absence of a generative model providing insights into causal relationships between the modular organization, neuronal hubs and segregation/integration, the authors should clear state that their discussion of mechanisms is primarily speculative, and suggest strategies for gaining further mechanistic insight.

Thank you for your comment.

Indeed, both the generative network model and neural circuit model are important approaches to provide us with evidence of the hypothesis raised about the formation of effective networks. However, our aim of using a model was exclusive to testing the accuracy of the methodology applied. Because of this, we choose a simpler model considering only synapses weight and delay to generate dynamics. The use of the referred models can be addressed in further work.

In this way, we agree that the speculative character of our arguments should be clearly stated. Thus, we inserted the following paragraph in the "Limitations" section of the Discussion, lines 663-670.

“We formulated a hypothesis involving silent synapse activation and STDP mechanisms that might have a role in the activity-dependent self-organization of neural circuits. However, to provide insights into causal relationships between such hypothesis and integrations/segregation, modular organization, and important nodes, it would be interesting to construct a computational model showing the dependence of connectivity probability on Euclidean distance and that connections are established according to the synchronization of neurons. Another option is to apply electrical stimulation in neuronal cultures to induce control of the mentioned mechanisms.**”**

9) Effective connectivity is inferred by using a transfer entropy analysis of electrophysiological signals. However, it has been questioned that the transfer entropy (James, et al., 2016) does not quantify the flow of information as commonly assumed. Given this concern, some clarifications or comparisons with state-of-art methods need to be provided.

Thank you for this comment and for the opportunity to clarify it.

Indeed, we were very aware of these issues about information flow quantification pointed out by James et al. (2016). Actually, exactly because of these points we constructed a very rigorous pipeline, and we tested the methodology in a model (including that the model was sub-sampled, and many indirect connections could be overestimated, similar to what happens in MEA recordings). In order to clarify these points, we have added the following paragraph to the section “Validation of effective connectivity” in Results, lines 163-189:

“In our analysis, since we have access only to neuron spiking activity, the term “connections” reflects strong paths for information flow rather than actual structural connections (Lizier and Prokopenko, 2010). This type of inference in neural systems should be done carefully because of the number of influences that the analysis can have. Additionally, the relationship between the structural and effective connections may not fully match. Since the predicted transfer is computed from spike trains, not all possible dynamical states are presented in the spontaneous activity. With inhibition active, several communication paths may be inactive or silent, although structural connections may exist (Park and Friston, 2013). James, Barnett, and Crutchfield (2016) pointed out some limitations of TE in detecting information flow, they showed through simple examples how TE can overestimate flow and underestimate the influence. On the other side Garofalo et al. (2009), Ito et al. (2011) and Orlandi et al. (2014) showed how using a combination of time delays and adequate binning may overcome these limitations and make TE analysis a powerful tool for inferring effective connections. Furthermore, Nigam et al. (2016b) discussed the importance of handling firing rate and population bursts influence, as well as spurious connections caused by common drive and transitivity to have a more accurate effective connections inference.

Underpinned by these works we constructed the present methodology. To test the accuracy of the effective connectivity inference we used Izhikevich’s network model (Izhikevich, 2006). The network dynamics was constructed based on the synaptic weights and delays between neuron activity propagation. The model reflects information flow pathways based on structural connections between cortical excitatory and inhibitory neurons. The mean firing rate of excitatory neurons was 4.78 ± 0.84 Hz (mean ± SD) and of inhibitory neurons was 16.82 ± 2.19 Hz (mean ± SD). This model helps us to test the ability of the pipeline in detecting the predictive information transfer. Since we compared the transferred information computation with the structural couplings of neurons that generated the information transfer.”

10) What is the relation between the log-normal distribution of coupling weights and that of firing rate? What do these heavy-tailed distributions mean for collective network states and whether they are related to the key connectivity properties such as neuronal hubs? Given that all these phenomena have been reported in previous studies, without an in-depth investigation of the problems, the novel contribution of this study is somewhat unclear to me.

Thank you for your point.

Indeed, all those are still open questions that must receive attention in the future network neuroscience works to increase our knowledge in the understanding of brain networks.

Our intention in reporting these same results already reported in other studies is not to provide novelty but show the redundancy of results even when considering individual neurons as fundamental elements of the networks. In principle, we reported these distributions exclusively to show that the networks analyzed are comparative to those already reported in previous studies.

Despite all these questions being very pertinent and deserve a closer focus, they were not part of the scope of this work, which was to look at the development of effective networks from neuronal assemblies’ activity-dependent self-organization.

In this way, the novel contribution of our study relies on the fine-scale analysis of the formation of the information flow in neuronal assemblies. Which could be compared with neuronal firing rate and neuron physical location. Such a comparison is not possible when considering connections between neuron populations as in previous works. We elaborated on one hypothesis related to the activation of silent synapses as a mechanism related to this phenomenon. Different from others works we point out actual biological processes that could be related to the network topology formation. Bridging a sometimes existent gap between theoretical and experimental neuroscience. In this way, our results can help other scientists guide their research to look at synapses activation, GABAR switch, formation of effective networks in low [ca^2+^]_E_, intrinsic neuronal mechanisms related to firing rate control, etc.

11) In Figure 1c and Figure 1d, the log normal distribution is used to fit the data. Is this distribution better than other heavy-tailed distribution such as truncated Pareto distribution?

Thank you for your questions.

To fit the data with a truncated Pareto distribution we need to show the raw computed values and not the logarithm scale. What makes them more grouped and harder to see. In this way, to present the data more clearly we maintained the distributions with the Gaussian fit as also presented by Nigam et al. (2016, doi:10.1523/JNEUROSCI.217715.2016) and we included insets in the Figure 1c,d showing the distributions fitted by the generalized Pareto distribution.

12) Line 88 – "the term neuronal network to refer to the neuronal assemblies coupled to electrodes and effective networks to refer to the network inferred from the neural recordings". This sentence is not correctly highlighting these two central concepts of the paper. Considering what was written a few lines above, and by also looking at the Results and Methods sections, maybe it would be easier for the reader to follow something along these lines: "the term neuronal networks to refer to transiently coupled neuronal assemblies and effective networks to refer to the circuits inferred from neural recordings". Of course, this is just a suggestion. The aim is to make as clear as possible the intentions of the authors to disentangle a dynamical aspect ('neuronal assembly') and a structural aspect ('effective network').

Thank you very much for your comment and suggestion.

We have concluded that our intention in separating these two terms did not work as we thought. Here, the term “effective connectivity” reflects strong paths for information flow rather than actual structural connections and we suggest synaptic/neuron mechanisms that could be related to the development of these paths. However, we are restricted by the limitations of the connection inference technique, which uses the dynamics of the neuronal assemblies being formed. In this way, although we agree with the reviewer that it could sound better, we believe the term “neuronal network” would not work as desirable since we are referring to the neuronal assemblies, thus we decided to remove it from all the text.

13) Line 145 – "Only some connections extended for long distances, and these connections were more likely to also have a weight close to the geometric mean. Connections with lower and higher weight values were more likely to have shorter lengths." Very interesting! Can the authors provide a statistical characterization of these effects and a corresponding figure? In particular it is important to test if this feature is also present in the two other models (Watts-Strogatz and Kwok et al.). If yes, it should be mentioned. If not, then it is an important distinguishing feature, and in this case, this aspect should be considerably expanded.

Thank you for your comments.

We explored the statistical characterization and expanded the Figure 1g to Figure 2 and Figure 2 —figure supplement 1. Both Watts and Strogatz and Kwok et al. studied only the topological properties of the networks in their models. We did not use these models in our analysis, so it would be considerably difficult to test if these characteristics are also present in them. In fact, just a few works compared functional/effective connections and the physical distance between elements, which makes comparisons difficult. Since we do not know what originates these phenomena is difficult to expand the discussion, our considerations about these points are presented in the lines 441-445 in the Discussion section.

14) Line 290 – "the higher the neuron score, the higher the median firing rate over all DIVs." The authors took care of choosing the best inferring method available (transfer entropy). However, a big issue of the structure-from-dynamic approach is its circularity. Dynamical information is used to infer the structure of a network and to draw conclusions concerning how the dynamically-derived structure affects network dynamic itself. Therefore the correlation between firing rate and neuron 'hub' scoring (sentence at the beginning of this paragraph) more than being a result of the study might be a side-effect of the analysis approach. The authors already took the care of searching the available literature to find support (like Sung et al. 2005 for the firing rates of neurons), they are invited to expand their discussion on this point, and suggest experimental and analytical solutions, e.g. use electron microscopy to validate (at least a portion of) the activity-derived networks, or the literature they found to also compare their motifs results, or, in case, why their results cannot be compared with the available literature and electron microscopy data. The use of a spiking network to test the validity of the inference procedure is interesting and valid as a preliminary test, but would require an entire separate work to be able to use it to break the structure-from-dynamic circularity. So this point too needs to be clearly discussed.

Thank you for your comments and suggestions.

We believe there might be a misunderstanding or a miswriting at some points. Just to clarify, we did not intend to infer structure from dynamics. We believe this confusion was promoted by some bad choices for some key terms.

In our analysis ‘connections’ reflect strong paths for information flow rather than actual structural connections. To make the text clearer we removed the distinction between neuronal networks and effective networks, and we also provided a more detailed explanation of the model used to test the accuracy of effective connectivity analysis. Additionally, we tried to define in a better way what connections mean in our context. We believe in this way the apparent argument circularity will disappear. Thank you for pointing this out.

Nevertheless, indeed, the firing rate might influence the measures used to compute scoring analysis. Mainly by the super estimation of TE results. However, we controlled the firing rate influence in the construction of the networks by comparing TE measured from spike trains in the actual data to TE measured from jittered spike trains, and we also normalized the TE results by the entropy of the receiver neuron. In this way, we believe the influence of the firing rate was softened.

15) Line 350 – Even though it is the discussion, it should be clearly indicated that by "self-organization", in the context of the experimental strategy and presented results, the authors cannot mean functional connectivity (as it is hinted by the first paragraph, line 352-257), but only self-organization as self-reinforcement of synaptic connections between neurons with correlated firing (as it is done in the following paragraphs of discussion). If, for example, spatially organized stimuli through the electrode were used, and effects on the network structure were observed, it could have been possible to advance the idea that the network self-organization would be functional. In any case, the silent synapses hypothesis they suggest is fundamental to understand also functional results. They could maybe move the first paragraph to the end?

Thank you for your comments and suggestions.

We moved the first paragraph to the end, and we also discussed functional connectivity and the use of an evoked activity to better correlate neuronal structure and function, as shown below. Alterations can be found in lines 450-462 of the manuscript.

“Although this work focused on the effective connectivity during the selforganization of neuronal assemblies, the relationship between structural and functional/effective connectivity has been a subject of interest in many other works (Meier et al., 2016; Segall et al., 2012; Suárez et al., 2020). To relate structural and effective connectivity is not an easy task since inhibitory activities do not allow all possible communication paths to be explored when effective connectivity is inferred from neuronal dynamics. As pointed out by Park and Friston (2013) structural networks constrain functional networks but also many patterns of functional connectivity can emerge from fixed structural ones and give rise to high-level neurocognitive functions. In this way, the silent synapses hypothesis might also explain functional connectivity results. However, the use of spatially organized electrical stimuli through the electrodes could support a better way to establish a relationship between neuronal structure and function by evoked activity (Bauer et al., 2018).”

16) Line 62 "The spontaneous emergence of spatio-temporal patterns driven by neuronal activity is characteristic of a self-organizing system." maybe could be a little expanded given that the rest of the article will be a balance of structure and dynamic running over it.

Thank you for your suggestion. We inserted the following sentence in lines 64-71:

“Synapses promote the structural and functional coupling of neurons, by allowing the propagation of biochemical and electric signals (Südhof, 2021). However, to produce a significant post-synaptic effect the pre-synaptic stimuli need to follow some specific temporal and spatial requirements (Magee, 2000). Adaptative mechanisms regulate synapses by promoting cooperation and competition among them (Zhang et al., 1998). In this way, the maturation of synapses induced by neuronal spontaneous activity has an important role in the emergence of spatial and temporal patterns related to selforganizing systems.”

17) Line 75 "DIV" is first used without being defined. The definition (days in vitro) comes for the first time in the caption of figure 3 (line 1211)

Indeed, thank you for this observation. We corrected it and now we defined DIV in the Introduction section the first time the word was mentioned. We also take off all the “s” used together with the acronym DIV, once it already means “days” in vitro.

18) Line 194, "only 5 of the 13 patterns (5, 8, 11, 12, and 13), were", the comma after the right bracket should be dropped.

Thank you. The comma was dropped.

19) Line 216 "Subgraphs", "modules", and "communities" seem to be used interchangeably. However they underlie different meanings (respectively theoretical, architectural, and functional). Wouldn't it be easier for the reader to stick to one term, if just one meaning is intended, or explain the term and its use, where appropriate?The clustering grows towards the 12 DIV then decreases, as path length and small-worldness (Figure 2cde). Is this backed up by other data? We are in use with the idea that brain networks are small-world, it looks like these networks, towards the 20 DIVs are not.

Thank you for this comment and observation.

We agree that it is clearer for the reader to stick to one term. In this way, as we are studying the topology of effective networks, we pick the term “modules” to refer to the architecture of the networks. The terms “subgraphs” and “communities” were changed in all the text.

From an experimental perspective, we did not test over DIV, but looking at the boxplots of the clustering coefficient, we can notice a small fall after 12 DIV. However, this fall may be a result of the increase in physical or chemical environment changes. Neurons’ physiology in culture is very susceptible to variations in humidity, temperature, pH, osmolarity, etc. While the maturation of synapses is susceptible to cell density, number of glial cells (whose proliferation rates are high), etc. It is harder to maintain all these variables the same over time than start with the same conditions for every culture. That might explain the increase in the result variances. However, despite the variance, the results after 12 DIV for the small-worldness coefficient are still deemed significant when compared with random networks. Showing that, the small-world organization is present. From a theoretical perspective, it is hard for us to imagine that firstly the effective networks would develop a small-world architecture, and then for an interval of time in development they would develop another architecture and after this interval recover the small-world properties. Downes et al. (2012) suggested that the small-world topology emerges from a random one, we discussed this hypothesis in the Discussion section. Even though we propose a different conjecture, one architecture switch seems more reasonable than two. A better option to address these questions would be to study the development of the network topology by tracking which neuron is represented by each node. That would make it possible closely watch the formation of connections and information flow. However, this kind of study is still very difficult to conduct given the available technologies.

20) Line 290, "Figure 4d shows that the higher…". It should be "Figure 4e". Additionally, for clarity in the presentation of results, panel 4f (participation coefficient) should go above panel 4e (firing rate).

Thank you. Both alterations were made.

21) Line 346 "neuron loss drastically impacted the topological organization of the effective networks, resulting in a breakup of the networks into different components" In addition (and before) pathological conditions, the neuron loss can also be a (functional) feature of the multi-stage process of circuit refinement, as the authors found in the formation of communities.

Yes, we agree. We extended the discussion in lines 374-385 as presented below.

“Conversely, after 30 DIV, the networks had a change in topology, ultimately returning to an architecture similar to a random network. As we used recordings from neurons in culture, this effect might be explained by neuronal death associated with the difficulties of maintaining a healthy cell culture beyond 4 weeks (Kaech and Banker, 2006). Indeed, Figure 1b shows a decrease in the number of recorded neurons, although no significant decrease in the number of network edges has been detected (Figure 2a). The impact on the topology after such a period includes a breakup of the networks into different components, as can be seen in Figure 2b. This result can also indicate how the indiscriminate loss of neurons may drastically affect neural circuits during pathological conditions. However, the silencing of neurons may be a result of the multi-stage process of circuit refinement during development. To test this hypothesis novel in vivo studies should be performed.”

22) Lines 428,429 There is a list of properties but is written in a series of sentences (lacking verbs). "… Strogatz, 2001). A short path …" should be ".. Strogatz, 2001), a short path …". One line below, "… Ottino, 2004). And the presence …" should be: "… Ottino, 2004), and the presence …"

Thank you. The paragraph was corrected.

23) Line 774, the fonts for variables and text are the same, making difficult to read the equation.

Thank you for the observation. The equation was corrected.